# The role of internet use in the relationship between occupational status and depression

**Yujie Zhang**⭕*

School of International and Public Affairs, Shanghai Jiao Tong University, Shanghai, China

* zhangyujie@sjtu.edu.cn

## Abstract

### Background

The emerging information and communications technology (ICT) and society 5.0 have integrated digital innovation and imaginative creativity to solve social problems and create social value in an aging society. Especially in China, the penetration rate of internet use has become more prominent. Nevertheless, the role of internet use in the relationship between occupational status and depression remains unknown. The purpose of this study was to estimate how internet use moderates the relationship between occupational status and depression in a sample of older adults in China.

### Methods

A cross-sectional design was used to assess the relationship between occupational status, internet use and depression. This study employed data from the Chinese General Social Survey (CGSS) conducted in 2017. The full sample size of this study is 2,403. OLS regression was adopted to explore the correlation between occupational status and depression and the moderating role of internet use. Meanwhile, a heterogeneity analysis based on residence registration was implemented to explore the difference between rural and urban sample groups.

### Results

The findings suggested that higher occupational status was related to lower depression level. By playing an inhibitory moderating role between the association of occupational status and depression, internet use and occupational status formed a substituting effect. Meanwhile, the above moderating effect was only significant in urban group and insignificant in rural group. Besides, being male and having higher social class were significantly and negatively correlated with depression.

### Conclusion

This study began with the theory of social stratification and social mobility, added the theory of social capital, constructed an analytical framework of regulatory effect, put forward two basic assumptions, selected measurement indicators taking into account Chinese cultural

**Data Availability Statement:** The data underlying the results presented in the study are available in the Harvard Dataverse repository: https://doi.org/10.7910/DVN/SZUSBS.

**Funding:** The author received no specific funding for this work.

**Competing interests:** The author has declared that no competing interests exist.

factors, and conducted empirical tests using authoritative statistical software and national representative data, providing a new theoretical contribution to our understanding of the impact of occupational status and depression in developing countries.

## Introduction

In most societies, social members are often divided into different classes formally or informally due to their different characteristics such as power [1], property [2], education [3], family [4], race [5], gender [6], age [7], and occupation [8], that is, there is a system that gives different social members different social status. Occupational stratification is a very key feature to distinguish social members [9, 10].

In today's industrialized society, occupational status often becomes an indicator of a person's status in society and affects people's class mobility [11]. Therefore, the change of occupational status has also become an important indicator to predict the direction and degree of social class change. Past studies have shown that social class change is related to depression [12, 13]. Entering an aging society, older population is a highly vulnerable group to depression [14]. It is very important to understand the mechanism of how occupational status associates with depression and the role of internet use plays between these two variables. This exploration not only helps to promote the theoretical research on social stratification and social mobility, but also has practical significance. Especially in the context of aging and information society, it helps to clarify the main obstacles to promoting the mental health of older adults and the direction of future efforts.

This study focuses on two dimensions of occupational status: social stratification and social mobility. The former refers to the stratification phenomenon of social members due to different possession of social resources [15], which can be regarded as the root cause of depression symptoms. The latter refers to the upward, downward or horizontal mobility of individuals in the social class [16]. Usually, social class is the main factor that can directly affect mental health [17], similar to proximate causes. Based on the review of existing literature and the theoretical analysis of social stratification and social mobility, this paper proposes that there is a positive relationship between occupational status and depression. Since entering the society 5.0 [18], where information technology has changed people's way of life, internet use has a moderating effect between occupational status and depression. More frequent internet use can significantly weaken the relationship between occupational status and depression. The empirical part focuses on China, using a national cross-sectional data set, and using statistical analysis methods to test the theoretical assumptions.

There are three main points of view about the impact of occupational status on depression. One view is that occupational status can promote mental health and alleviate depressive symptoms. Diaz et al. believed that low subjective occupational status perception was an important factor in predicting depression symptoms [19]. Murphy et al. found that the prevalence of depression in people with low occupational status was significantly higher than that in people with high occupational status. There is also a trend that the downward social mobility predicted by low occupational status is related to depression. This finding supports the view that depressed people are concentrated at the low end of the social class [20]. Singh-Manoux et al. analyzed the prospective cohort study of civil servants in London. It shows that subjective occupational status perception is a powerful predictor of poor mental health.

Subjective occupational status reflects the average cognitive level of socio-economic status, without psychological bias [21]. In this regard, common explanations include that the social

stratification system will produce differences in mental health [22], and the sense of unfairness caused by poor social mobility [23]. Generally speaking, this kind of view is mainly used to discuss the occupational status of western developed countries. The main characteristics of western developed countries are that the economic development speed slows down, the mobility of social classes decreases, the social class is relatively solidified, it is very difficult for the bottom to squeeze into the middle class, and it is more difficult for the middle class to squeeze into the top [24–26]. In developing countries like China, the economic development is in the stage of rapid and steady growth, and the proportion of the middle class is still at a relatively low level [27]. In the future, a considerable number of rural families or working families will squeeze into the middle class through efforts and ability. Some middle-class families with strong ability can even squeeze into the rich class. Nevertheless, this involves efforts, capabilities, networks and other aspects [28].

Another view is that occupational status will aggravate psychological stress and aggravate depressive symptoms. Demerouti et al. and Seto et al. used the methods of causal analysis of longitudinal data sets and correlation analysis of cross-sectional data sets to test the negative effects of work stress and family career conflict caused by high occupational status, and found that high occupational status can lead to the aggravation of depression, and work stress and fatigue are the determinants of depression; Moreover, as time goes on, work stress and fatigue have a causal relationship and a reverse causal relationship, thus forming a vicious circle [29, 30]. Song's comparative study and analysis of China and the United States found that, unlike the positive effect of high occupational status on mental health in the United States, in Chinese cities, the upward (relative to downward) social comparison motivation is stronger than that in the American society with a strong individualist culture. High occupational status also means facing higher peer pressures, which may cause individuals to feel dissatisfied with their own situation, thus aggravating depression symptoms [31].

The third view is that the relationship between occupational status and depression is regulated by other factors. For example, Nishimura pointed out that although it was found in the comparative study of Japan, South Korea and China that occupational status had an impact on depression, the investigation of the labor market structure of three different societies showed that the impact of occupational status was regulated by other variables, such as social culture and institutional framework. High occupational status can reduce depression symptoms through two distinct mechanisms: one is to provide sufficient financial stability and provide employees with additional benefits and other potential economic resources; the other is to avoid employees from experiencing various stressors corresponding to low occupational status, such as economic stress and work overload. In China, for example, people working in private enterprises have a higher level of depression than those working in the party and government departments, partly because of the greater economic pressure and work overload in private enterprises [32]. In addition, even for older adults, the soundness of the social welfare system also plays a regulatory role: for example, for low-income countries and middle-income countries, in China, where the soundness of social welfare is relatively strong, high occupational status means higher pensions, which to a large extent can alleviate the depressive symptoms of older adults; However, in India, where the social welfare system is relatively unsound, the dependence between social welfare and occupational status is weak, and it is difficult for occupational status to alleviate depressive symptoms through the welfare mechanism [33].

Based on the existing literature, this paper attempts to test the relationship between occupational status and depression. This paper mainly focuses on the occupational status of developing countries, and its association with depression is significantly different from that of western developed countries. First, the social stratification in western developed countries is relatively

fixed, and occupation can be used as a representative standard for stratification. In developing countries, social mobility is fast, and the types of emerging occupations continue to iterate. The correlation between occupation status and depression may be weaker than that in western developed countries. Second, for developing countries, the rapid urbanization process has widened the gap between urban and rural areas. Because the rural vocational system is single, mainly focusing on agriculture, while the urban vocational system is diverse, forming a differentiated hierarchical system. Therefore, the association between occupational status and depression has urban-rural heterogeneity, which is more significant in cities. Third, different from the Chinese society before the reform and opening up, since the reform and opening up in 1978, the Chinese government has gradually established a modern social security system. For older population, high occupational status also means higher pension and more stable welfare security, which will promote their mental health after retirement [34]. Therefore, this study proposes the following hypothesis.

Hypothesis 1: Occupational status is negatively associated with depression.

The relevant literature discusses the relationship between internet use, mental health and occupational status. On the one hand, scholars generally recognize that internet use is conducive to improving depression. For example, Liu et al. proposed that internet use is conducive to promoting social participation and has important value in maintaining mental health [35]; On the other hand, König et al. found that in Europe, the internet use of older adults is driven by the impact of previous work experience. Behind the differences in occupational status is the impact of social inequality and different social and economic resources on older population's access to new technologies for internet use [36]. At the same time, older adults with low internet use skills are increasingly affected by the negative impact of social stratification. In the view of some scholars, it is necessary to focus on the social 5.0 perspective and help the older adults improve their mental health by providing internet use training and other means to resist the negative impact of social stratum consolidation and poor mobility [37–39].

In recent years, more scholars have tested the role of internet use in social stratification and social mobility based on empirical research design. Eynon et al. based on the analysis of four waves of longitudinal data from the United Kingdom, found that internet use helps to improve social mobility, and this effect is more significant in western developed countries and the promotion of social stratification represented by occupational status [40]. Kappeler et al. took how the digital divide in Switzerland—social differences in Internet adoption—evolved from 2011 to 2019 as an example, and found that the use of the Internet was still stratified according to existing social stratifications, while the situation of not using the Internet was increasingly concentrated in traditionally disadvantaged occupations [41]. Yoon et al. found that for older adults, the lower level of occupational status significantly reduced the probability of using the Internet to obtain health information [42]. Barbosa Neves et al. proposed that the Internet seems to help maintain, accumulate and even mobilize social capital. Moreover, it also seems to exacerbate social inequality and cumulative advantage [43]. Based on Max Weber's political perspective, Blank and Groselj pointed out that the main sources of social stratification are social class, occupational status and power. As the Internet becomes more and more important, it has steadily risen to a more central position in the hierarchical system. Therefore, it is important to look at the use of the Internet from the perspective of Marx Weber, and explore how social class, occupational status and power help explain people's Internet realization [44]. In addition, although recent studies have shown that the scale of the digital divide has been shrinking for many groups, there is a strong correlation between socio-economic factors such as occupational status and internet use [45].

Based on the literature, this study further points out that internet use plays an important role in reducing the relationship between occupational status and depression. The existing

literature mainly discusses the impact of internet use from the perspective of social stratification and social mobility. However, the association between occupational status and depression of older adults is long-term and has accumulated for many years. Some older adults have even retired, so it is difficult to change their occupational status. Will the use of the Internet help ease this long-term relationship? At present, there is still a lack of detailed demonstration on this issue, but this paper believes that the answer is yes. Internet use has a positive effect on alleviating the long-term relationship between occupational status and depression.

Internet use is a multi-dimensional concept. This study focuses on the role of internet use frequency. The frequency of internet use has a direct impact on mental health, which determines whether individuals can effectively participate in society and accumulate social capital in the information age. It is a direct factor affecting mental health. Choi and DiNitto found that internet use can establish social integration and support networks for older adults, and enhance psychological capital, namely emotional well-being and self-efficacy, based on a representative sample of the older population aged 65 and over in the United States [46]. Older people who use the Internet more often usually report better self-rated health [47]. According to Matthews et al., for the older adults with lower socio-economic status, the increase of internet use frequency has greater health benefits than those with higher socio-economic status [48]. In the long run, because the use of the Internet has improved social capital, it has also reduced the older population's concern that their social and economic status is inferior to others to a certain extent [49]. In reality, the low occupational status often restricts the improvement of the economic ability of older adults, so that too many resources benefit the people with high occupational status. The living conditions of people with low occupational status are relatively poor, which is not conducive to mental health [50]. However, the increase of social capital caused by internet use improves the social support that older adults can obtain. Social support is conducive to mental health [51]. Therefore, internet use is an inhibitory intermediary factor between occupational status and depression. This study proposes the following hypothesis.

Hypothesis 2: Internet use is an inhibitory moderator between occupational status and depression.

To the best of the author's knowledge and based on this research, there were no previous studies concerning the relationship between occupational status, internet use and depression among older adults in China. This study was conducted to fulfill this research gap. The purpose of this study was to estimate the moderating role of internet use on the association between occupational status and depression in society 5.0.

## Methods

### Data and sample

The cross-sectional data of this study were obtained from the 2017 wave of the Chinese General Social Survey (https://doi.org/10.7910/DVN/SZUSBS). The CGSS was conducted by the Survey and Data Center of Renmin University of China. At present, CGSS data has become the main data source for the study of Chinese society and is widely used in scientific research, teaching, and government decision-making. CGSS adopted stratified multiple-stage probability sampling method in the survey, and was carried out in all urban and rural households in 31 provinces, autonomous regions, and municipalities directly under the central government (excluding Hong Kong, Macao, and Taiwan), which makes CFPS data representative and authoritative with scientific research value. Moreover, CFPS was implemented by a group of trained researchers through face-to-face interviews, later family visits, and telephone surveys, which ensures the high quality of the data.

The study did not include minors. The researchers obtained the written informed consent from the respondents. CGSS was conducted following the ethical principles in the Declaration of Helsinki. After ensuring that the potential respondents understood the information. Each potential respondent was fully informed of the research purpose, method, funding source, any possible conflict of interest, institutional affiliation of the researchers, expected benefits and potential risks of the research, possible discomfort caused by the research, and any other information related to the research.

The sample size of this project was 12,582. The subjects of this study were older adults over 60 years old. After the screening and elimination of samples lacking relevant variables, 2,403 effective samples were obtained.

Because the present study was conducted based on the de-identified, publicly available CGSS data, it does not constitute human subject research. Its institutional review board review was waived because there was no interaction with any individual, and no identifiable private information was used.

## Measures

In this study, three core variables (i.e. occupational status, internet use and depression) were measured using the Likert 5-point scale. The Likert 5-point scale consists of a group of statements. Each statement has five responses, namely "strongly agree", "agree", "not necessarily", "disagree" and "strongly disagree", which are recorded as 5, 4, 3, 2 and 1 respectively. What is measured is the attitude or status of each respondent on the scale [52]. The Likert scale usually uses five response levels, but many psychometrists advocate using more levels [53]. However, a recent empirical study pointed out that after simple data conversion, the average, variance, skewness and kurtosis of the data with 5-level, 6-level and 11-level options are very similar, so statistically speaking, there is little difference [54]. In addition, although the literature shows that the Likert scale with more levels of options is more sensitive. Validity may also improve [55]. However, Chinese people are not very sensitive to the use of words, and some words are easy to confuse the respondents [56]. For example, "some agree" and "a little agree", some respondents reported that it is difficult to distinguish. In addition, the subjects of this study are older adults, who do not have a strong ability to distinguish the details of language words [57]. Therefore, the Likert 5-point scale is more suitable to measure their state.

**Depression.** In CGSS, respondents were asked about their self-reported feeling of depression as follows: "How often have you felt depressed or frustrated in the past four weeks?" This item was scored on a 5-point Likert scale ranging from 1(never) to 5(always).

**Occupational status.** Research of occupational status in China suggested that in people's subjective cognition, the Communist Party of China (CPC) and the government has the highest occupational status, enterprise takes the second place [58]. As there were wage restrictions for nonprofits and grass-roots organizations in China, and their legitimacy depended on relationships with the government, public institutions affiliated to government had higher occupational status compared with nonprofits or grass-roots organizations [59]. Since there was no social security or social recognition for unemployed individuals, the occupational status of unemployment was the lowest [60]. In CGSS, respondents were asked about their occupational status, namely, "What is your occupational status type?" ("unemployment" was coded as 1, "nonprofits or grass-roots organizations" was coded as 2, "public institutions" was coded as 3, "enterprise" was coded as 4, and "the Communist Party of China and the government" was coded as 5). In general, the higher the score, the higher the occupational status is.

**Internet use.** In CGSS, respondents were asked about their internet use frequency, namely, "In the past year, do you often engage in online activities in your spare time?"

("never" was coded as 1, "several times a year or less" was coded as 2, "several times a month" was coded as 3, "several times a week" was coded as 4, and "everyday" was coded as 5).

**Covariates.** Based on the existing literature [55–58], age (continuous variable), sex (0 = "female", 1 = "male"), education (1 = "below junior high school", 2 = "above junior high school, and below undergraduate", 3 = "above undergraduate"), income (1 = "below RMB 50,000 per year", 2 = "above RMB 50,000, and below RMB 100,000 per year", 3 = "above RMB 100,000 per year"), social class (1–10, with a higher score indicating a higher social class) and residence registration (0 = "rural", 1 = "urban") were selected as control variables.

## Statistical analysis

All the statistical analyses were performed with Stata version 16.0 (StataCorp, Texas of United States). Stata is a statistical program developed by StataCorp in 1985. It is widely used in enterprises and academic institutions. Many users work in research fields, especially in economics, sociology, politics and epidemiology. Stata has a strong statistical function. In addition to the traditional statistical analysis methods, it also collects new methods developed in recent 20 years, such as Cox proportional risk regression, exponential and Weibull regression, logistic regression between multi category results and ordered results, Poisson regression, negative binomial regression and generalized negative binomial regression, random effect model, etc. Specifically, Stata has the following statistical analysis capabilities: general analysis of numerical variable data: parameter estimation, one-way and multi factor variance analysis, covariance analysis, interaction effect model, balanced and unbalanced design, nested design, random effect, pairwise comparison of multiple means, processing of missing data, variance homogeneity test, normality test, etc.

The statistical analysis is conducted in four stages. First, this study conducted a descriptive analysis of the variables. Next, based on an OLS model, a moderating effect analysis was performed to test whether internet use moderated the association between occupational status and depression. Then, based on residence registration, a heterogeneity analysis was performed to test if occupational status moderated the association between occupational status and depression both in rural and urban groups. Lastly, a robustness test was performed by using the ordered logit regression. And the results were robust. In the present study, the Variance Inflation Factor (VIF) values < 10, which indicated that multicollinearity was not an issue in the estimate. Fig 1 shows the research framework.

## Results

### Descriptive statistics

Table 1 displays the descriptive statistics of all variables. The mean of depression score was 2.059, and the standard deviation was 1.021, which indicates that the respondents had relatively low depression level and that there was no large disparity within this variable. The mean of occupational status score was 3.398, which suggests that the overall occupational status was medium. The mean of internet use score was 2.317, which shows that the frequency of internet use among respondents was relatively low. The respondents aged from 60 to 106. The proportion of female adults and male adults in the sample was balanced. Most respondents had received junior high school education and had a yearly income between RMB 50,000 and RMB 100,000. The mean of social class score was 4.389, which implies that most respondents subjectively believed that they were in the lower middle class of society. Notably, most respondents' residence registered in urban.

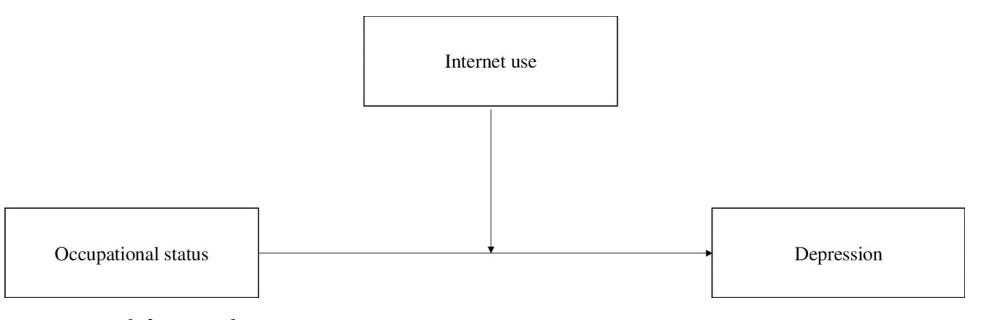

**Fig 1. Research framework.**

## Hierarchical regression analyses

Table 2 shows the results of hierarchical regression analyses. In the first step, occupational status was negatively correlated with depression (coefficient = -0.078, $p < 0.001$). Control variables including sex, education, social class, and residence registration significantly correlated with depression in a negative way. In the second step, occupational status was negatively correlated with depression (coefficient = -0.074, $p < 0.001$). Control variables including sex, social class and residence registration significantly correlated with depression in a negative way. In the third step, occupational status was negatively correlated with depression (coefficient = -0.126, $p < 0.001$). Internet use was negatively correlated with depression, too (coefficient = -0.132, $p < 0.01$). Notably, the interaction item of occupational status and internet use was positively correlated with depression (coefficient = 0.026, $p < 0.05$). Control variables explained depression in the same way with which in Step 2. It was suggested that internet use may play an inhibitory moderating role between occupational status and depression and further investigation should be conducted. In sum, the results supported Hypothesis 1.

## The moderating effect of internet use

Table 3 illustrates the moderating effect of internet use. After centralization of occupational status and internet use, the regression results revealed that internet use played an inhibitory role in moderating the association between occupational status and depression (coefficient = 0.026, $p < 0.05$). Moreover, sex, social class and residence registration were significantly negatively related to depression. Besides, the marginal effect of occupational status and internet use was significant and negative, which suggests that there may exit substituting effect between the two variables.

**Table 1. Descriptive statistics of respondents (N = 2,403).**

| Variable | Mean | SD | Min | Max | N |
|---|---|---|---|---|---|
| Depression | 2.059 | 1.021 | 1 | 5 | 2,403 |
| Occupational status | 3.398 | 1.101 | 1 | 5 | 2,403 |
| Internet Use | 2.317 | 1.756 | 1 | 5 | 2,403 |
| Age | 72.922 | 8.335 | 60 | 106 | 2,403 |
| Sex | 0.510 | 0.500 | 0 | 1 | 2,403 |
| Education | 1.749 | 0.535 | 1 | 3 | 2,403 |
| Income | 1.257 | 0.536 | 1 | 3 | 2,403 |
| Social class | 4.389 | 1.717 | 1 | 10 | 2,403 |
| Residence registration | 0.808 | 0.394 | 0 | 1 | 2,403 |

Table 2. Hierarchical regression results (N = 2,403).

| Variable | Step 1 | Step 2 | Step 3 |
|---|---|---|---|
| Occupational status | -0.078*** | -0.074*** | -0.126*** |
| | (-3.90) | (-3.74) | (-4.16) |
| Internet use | | -0.039** | -0.132** |
| | | (-3.02) | (-3.07) |
| Occupational status * Internet use | | | 0.026* |
| | | | (2.26) |
| Age | 0.0005 | -0.002 | -0.002 |
| | (0.19) | (-0.67) | (-0.60) |
| Sex | -0.136*** | -0.140*** | -0.140*** |
| | (-3.29) | (-3.37) | (-3.39) |
| Education | -0.092* | -0.065 | -0.065 |
| | (-2.23) | (-1.52) | (-1.54) |
| Income | -0.071 | -0.058 | -0.061 |
| | (-1.80) | (-1.48) | (-1.55) |
| Social class | -0.100*** | -0.098*** | -0.097*** |
| | (-8.30) | (-8.12) | (-8.04) |
| Residence registration | -0.312*** | -0.285*** | -0.273*** |
| | (-5.22) | (-4.73) | (-4.51) |

Notes: t values in parentheses

*** $p < 0.001$

** $p < 0.01$

* $p < 0.05$.

Fig 2 shows the inhibitory moderating effect of internet use in the relationship between occupational status and depression. This means that when older adults had higher level of internet use, the same occupational status score increase would lead to less depression reduction. Although occupational status was negatively associated with depression, there were the following conditions: the negative role of occupational status was relatively strong when internet use was low, but the negative role of occupational status gradually decreased with an increase in internet use. In sum, the results supported Hypothesis 2.

Table 3. Moderating effect analysis (N = 2,403).

| Variable | coefficient | t value | p > \|t\| |
|---|---|---|---|
| C_Occupational status | -0.065*** | -3.22 | 0.001 |
| C_Internet use | -0.042** | -3.27 | 0.001 |
| C_Occupational status * C_Internet use | 0.026* | 2.26 | 0.024 |
| Age | -0.002 | -0.60 | 0.548 |
| Sex | -0.140*** | -3.39 | 0.001 |
| Education | -0.065 | -1.54 | 0.124 |
| Income | -0.061 | -1.55 | 0.122 |
| Social class | -0.097*** | -8.04 | 0.000 |
| Residence registration | -0.273*** | -4.51 | 0.000 |

Notes

*** $p < 0.001$

** $p < 0.01$

* $p < 0.05$.

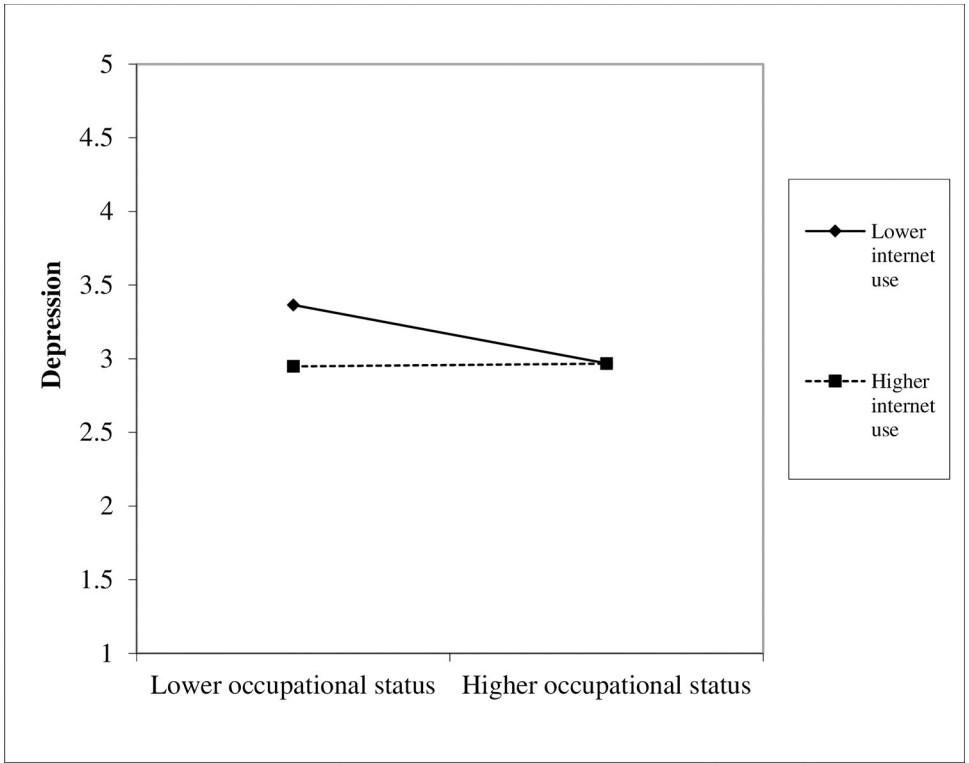

**Fig 2. Moderating effect of internet use.**

Table 4 shows the correlation between occupational status and internet use. This means that internet use was significantly positively correlated with occupational status (coefficient = 0.036, p < 0.01). Age and residence registration were two variables positively correlated with occupational status whereas sex, education, income and social class were not related to occupational status.

## Analysis of heterogeneity of residence registration

Table 5 shows the results for heterogeneity analysis. For urban group, the moderating effect of internet use was significant, and internet use had an inhibitory effect on the association

**Table 4. Correlation between occupational status and internet use (N = 2,403).**

| Variable | coefficient | t value | p > \|t\| |
|---|---|---|---|
| Internet use | 0.036** | 2.70 | 0.007 |
| Age | 0.008** | 2.99 | 0.003 |
| Sex | 0.058 | 1.36 | 0.174 |
| Education | 0.087 | 1.99 | 0.046 |
| Income | 0.007 | 0.18 | 0/856 |
| Social class | 0.015 | 1.22 | 0.223 |
| Residence registration | 1.018*** | 17.39 | 0.000 |

Notes: *** p < 0.001

** p < 0.01

* p < 0.05.

**Table 5. Residence registration heterogeneity analysis (N = 2,403).**

| Variable | By residence registration | |
|---|---|---|
| | Rural | Urban |
| C_Occupational status | -0.007 | -0.096*** |
| | (-0.16) | (-3.82) |
| C_Internet use | -0.030 | -0.046*** |
| | (-0.63) | (-3.31) |
| C_Occupational status * C_Internet use | 0.027 | 0.036* |
| | (0.97) | (2.52) |
| Age | -0.011 | 0.0002 |
| | (-1.65) | (0.08) |
| Sex | -0.184 | -0.134** |
| | (-1.74) | (-2.97) |
| Education | -0.234* | -0.030 |
| | (-2.38) | (-0.64) |
| Income | -0.220 | -0.047 |
| | (-1.91) | (-1.12) |
| Social class | -0.108*** | -0.095*** |
| | (-4.26) | (-6.92) |

Notes: t values in parentheses

*** $p < 0.001$

** $p < 0.01$

* $p < 0.05$.

between occupational status and depression. Meanwhile, control variables such as sex and social class were significantly and negatively correlated with depression in the urban group. For rural group, the moderating effect of internet use was insignificant. Except for education and social class, other control variables were not related to depression in the rural group.

## Robustness test

The robustness test examines the robustness of the interpretation ability of the evaluation methods and indicators, that is, whether the evaluation methods and indicators still maintain a relatively consistent and stable interpretation of the evaluation results when some parameters are changed. Starting from the measurement method, this study changes the OLS regression method originally used to the ordered logit regression model and carries out repeated experiments to observe whether the empirical results change with the change of measurement method. If the measurement method is changed, the sign and significance are found to change, indicating that it is not robust, and the problem needs to be found. Table 6 shows that after this study used the ordered logit regression model for regression analysis, overall, the results were consistent with previous OLS regression model, so the results were robust.

## Discussion

This study attempts to explain whether occupational status is related to depression and the regulatory role of internet use in the above relationship. The results showed that: in general, the occupational status was indeed negatively correlated with depression, and the depressive symptoms of people with higher occupational status were lighter; Internet use has a negative regulatory effect on the relationship between occupational status and depression. High

**Table 6. Results of robustness test (N = 2,403).**

| Variable | coefficient | z value | p > \|z\| |
|---|---|---|---|
| C_Occupational status | -0.108** | -2.84 | 0.004 |
| C_Internet use | -0.088*** | -3.55 | 0.000 |
| C_Occupational status * C_Internet use | 0.047* | 2.09 | 0.036 |
| Age | -0.004 | -0.82 | 0.410 |
| Sex | -0.238** | -3.05 | 0.002 |
| Education | -0.145 | -1.82 | 0.068 |
| Income | -0.121 | -1.62 | 0.106 |
| Social class | -0.168*** | -7.32 | 0.000 |
| Residence registration | -0.480*** | -4.32 | 0.000 |

Notes

*** p < 0.001

** p < 0.01

* p < 0.05.

frequency internet use helps to alleviate the negative correlation between occupational status and depression.

This study has clear policy implications for promoting mental health and alleviating the negative psychological effects of the solidification of social stratification and the slowdown of social mobility. First, we should make effective use of digital technology empowerment to bridge the digital divide and promote the construction of Internet skills training programs for older adults. Internet application is an important symbol of social 5.0 and an important way to accumulate social capital [43, 46]. Targeted promotion projects must be designed to bridge the long-term negative psychological effects caused by low professional status [22–25]. At present, there is a huge digital divide in society. Older adults with low professional status have weak internet use skills [44]. The realization of the positive role of the Internet needs to be based on sufficient infrastructure construction, that is, the government needs to strengthen the Internet infrastructure construction first. Therefore, we should give further play to the enabling role of digital technology in mental health, promote the construction of society 5.0 nationwide, especially in rural areas with low Internet penetration, and break the phenomenon of "digital divide" [42, 45]. At the same time, we should establish a smooth social flow channel and a sound social security system, so that even older adults with low professional status can have a better mental health status and get equal social service resources.

This study has several limitations. First, the study has limited generalizability. Although CGSS encompasses a large range of older adults, respondents in this study are above the age of 60. This study uses age 65 as the lower threshold for older adults even though 60 is relatively old in the context of life expectancies in developing countries. Therefore, setting the cutoff to an older age limited the sample too much for generalization of the conclusions in developing countries. Future research is needed to explore the relationship using nationally representative data for individuals before age 60.

Second, this study was not able to examine reasons for the selection of occupation and how those factors might interact with mental health of older adults. Factors that related to the selection of occupations (e.g., social connections, technical skills) may impact mental health differently, and those effects may differ by occupational status of older adults. Therefore, qualitative studies of this topic could be carried out to reveal the nuanced mechanism that links occupational status and depression.

Finally, the results were cross-sectional; thus, they only examined the relationship between occupational status, internet use, and depression. Longitudinal and experimental studies should be conducted to reveal the causal relationship among the above variables.

The strengths of this article are related to the straightforward measurement of occupational status, internet use and depression using Likert 5-point scale and the large number of respondents that are studied. First, the way three core variables are measured is an improvement upon many other studies. In line with theoretical considerations of the applicability, three core variables are measured by questions regarding the state of the respondents [52, 53]. Second, based on the cultural background of China, this study measured three core variables using 5 levels of Likert scale [56], which resulted in reliable estimations of the respondents' status. Third, this study systematically accounted for individual demographic and socio-economic conditions as well, while studying the relationship between occupational status, internet use and depression.

The findings here are in line with the earlier results of Choi and DiNitto, who also established a conceptual framework to investigate how internet use could accumulate social capital to alleviate depression of older adults by improving self-efficacy [46]. Although social stratification theory and social mobility theory already provide a good introduction to how occupational status is associated with depression [16–19], the perspective of social capital theory provides a new discovery for us to understand how internet use weakens this connection. In this case, this study began with the theory of social stratification and social mobility, added the theory of social capital, constructed an analytical framework of regulatory effect, put forward two basic assumptions, selected measurement indicators taking into account Chinese cultural factors, and conducted empirical tests using authoritative statistical software and national representative data, providing a new theoretical contribution to our understanding of the impact of occupational status and depression in developing countries.

## Conclusion

This study examines the correlation between occupational status and depression and the role of internet use in moderating the above relationship. This study found that occupational status was significantly negatively correlated with depression. An increase in internet use can inhibit the negative relationship between occupational status and depression.

## Author Contributions

**Conceptualization:** Yujie Zhang.

**Data curation:** Yujie Zhang.

**Formal analysis:** Yujie Zhang.

**Investigation:** Yujie Zhang.

**Methodology:** Yujie Zhang.

**Project administration:** Yujie Zhang.

**Resources:** Yujie Zhang.

**Software:** Yujie Zhang.

**Supervision:** Yujie Zhang.

**Validation:** Yujie Zhang.

**Visualization:** Yujie Zhang.

**Writing – original draft:** Yujie Zhang.

**Writing – review & editing:** Yujie Zhang.

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
