## [Decision Letter · Decision Letter 0]

28 Mar 2022

PONE-D-22-00051The relationship between work units' ownership form and mental health among older adults in China: The moderating role of internet usePLOS ONE

Dear Dr. Zhang,

Thank you for submitting your manuscript to PLOS ONE. After careful consideration, we feel that it has merit but does not fully meet PLOS ONE’s publication criteria as it currently stands. Therefore, we invite you to submit a revised version of the manuscript that addresses the points raised during the review process.

We look forward to receiving your revised manuscript.

Kind regards,

László Vasa, PhD

Academic Editor

PLOS ONE

Journal Requirements:

(This study was supported by Human Frontier Science Program grant (RGP0022/2014; www.hfsp.org) to L.C.; Engineering and Physical Sciences Research Council program grant Brains-on-Board (EP/P006094/1; epsrc.ukri.org) to L.C.; European Research Council grant SpaceRadarPollinator (339347; erc.europa.eu) to L.C.; and a Royal Society Wolfson Research Merit Award (royalsociety.org) to L.C. The funders had no role in study design, data collection and analysis, decision to publish, or preparation of the manuscript.)

(This study was supported by Human Frontier Science Program grant (RGP0022/2014; www.hfsp.org) to L.C.; Engineering and Physical Sciences Research Council program grant Brains-on-Board (EP/P006094/1; epsrc.ukri.org) to L.C.; European Research Council grant SpaceRadarPollinator (339347; erc.europa.eu) to L.C.; and a Royal Society Wolfson Research Merit Award (royalsociety.org) to L.C. The funders had no role in study design, data collection and analysis, decision to publish, or preparation of the manuscript.)

(This study was supported by Human Frontier Science Program grant (RGP0022/2014; www.hfsp.org) to L.C.; Engineering and Physical Sciences Research Council program grant Brains-on-Board (EP/P006094/1; epsrc.ukri.org) to L.C.; European Research Council grant SpaceRadarPollinator (339347; erc.europa.eu) to L.C.; and a Royal Society Wolfson Research Merit Award (royalsociety.org) to L.C. The funders had no role in study design, data collection and analysis, decision to publish, or preparation of the manuscript.)

Reviewers' comments:

Reviewer's Responses to Questions

**Comments to the Author**

1. Is the manuscript technically sound, and do the data support the conclusions?

Reviewer #1: No

Reviewer #2: Yes

Reviewer #3: Yes

2. Has the statistical analysis been performed appropriately and rigorously? 

Reviewer #1: I Don't Know

Reviewer #2: Yes

Reviewer #3: Yes

3. Have the authors made all data underlying the findings in their manuscript fully available?

Reviewer #1: No

Reviewer #2: Yes

Reviewer #3: Yes

4. Is the manuscript presented in an intelligible fashion and written in standard English?

Reviewer #1: No

Reviewer #2: Yes

Reviewer #3: Yes

5. Review Comments to the Author

Reviewer #1: The paper covers a promising topic, however, the realization is quite poor.

The study is too short and the structure isn't eligible for a scientific writing.

Abstract isn't clear, title is too complicated and incomprehensive. Actually, the whole paper is incomprehensible.

The literature review is totally missing, only very few sources were used for this study. Methodology is not clear, selection reasons aren't explained.

Based on the results, the evaluation is not possible.The role of internet use is not explained.

I recommend to rewrite it by quality standards and resubmit it again.

Reviewer #2: The paper presents the results of a methodologically sound, logically conducted study.

The objectives and results are clearly defined by the authors and the modelling of the research process is professional. The results and main conclusions are formulated.

This study not only investigated the effects of work unit’s ownership form on the mental health of the elderly, but also introduced internet use as a moderating variable to analyze the impact mechanism. With the acceleration of the digital process, this study once again proves that internet can bridge the mental health gap caused by the nature of the work unit.

Main conclusions are:

Work unit’s ownership form could significantly impact the mental health of the elderly.

Internet use inhibits the relationship between work unit’s ownership form and mental

health promotion.

Their findings suggest that work unit’s ownership form was significantly associated with mental health status for older Chinese residents, moderated by internet use.

While it is true that the study focuses on the specificity of China, I think that it will be worthwhile to look outside and make comparisons with other countries as a continuation of the research even if Chinese work units have unique attributes.

The link between the descriptive part of the study and the empirical phase would have been further strengthened by supporting the research background and the precise research objectives with literature and preliminary research results.

The study aims to explore how the nature of work units affects the mental health of the elderly, and to explore the regulatory role of Internet applications in the era of digital governance. The authors have met this objective within the scope of the study.

The limitations of the study are mentioned, and plans to overcome them should be shared with the reader in more detail, so that they can see more clearly the research model the authors intend to follow in pursuing data collection.

I consider the strength of the study to be its methodology and its very thorough and professional presentation. The reader is presented with a logical research process model and a presentation of results.

Even if the results are regional, I think that they can serve as a good example for further researchers and for further studies on similar topics. The results presented in this study have many relevant elements not only from a theoretical point of view but also from a practical point of view.

Reviewer #3: The topic of the paper is really interesting and in the face of the emerging novel ICT and society 5.0 extremely relevant as well. While the importance and effect of work units was emphasised and underlined with references, the internet usage and its potential relation with mental health was not addressed directly in the introductory/literature review part. The first mention of related literature can be found in the discussion part. What is more, the relation of work unit ownership form and internet use is not explored, even though they are suggested to have strong correlation according to CFPS.

The chosen methodology is in line with the research question and the aim and purpose of the research, however, the data presented in the paper could be enriched along the previous logic.

While author references 30 relevant international sources, only 3 from past 5 years, no form past 2, even though Covid19 has reshaped the economic and societal landscape of China as well. Author is recommend to read up on and refer to novel national and interantional literature especially realted to internet usage in different social and age groups.

6. PLOS authors have the option to publish the peer review history of their article (what does this mean?). If published, this will include your full peer review and any attached files.

Reviewer #1: No

Reviewer #2: **Yes: **Mónika Garai-Fodor

Reviewer #3: No

---

## [Author Response · Author response to Decision Letter 0]

17 May 2022

Dear Editor and Reviewers,

Thank you for the opportunity to resubmit my revised manuscript, “The role of internet use in the relationship between occupational status and depression”, for publication in PLoS One.

In the resubmission, I have carefully responded to the suggestions made by the editor and reviewers and modified the manuscript accordingly. I am very grateful for the comments, which assisted me in greatly improving the paper and enabled me to become more aware of issues of accuracy regarding the concept used, as well as the literature. I have also revised the English presentation of the full text to make the manuscript read more smoothly.

The emerging information and communications technology (ICT) and society 5.0 have integrated digital innovation and imaginative creativity to solve social problems and create social value in an aging society. Especially in China, the penetration rate of internet use has become more prominent. Nevertheless, the role of internet use in the relationship between occupational status and depression remains unknown. The purpose of this study was to estimate how internet use moderates the relationship between occupational status and depression in a sample of older adults in China.

The findings suggested that higher occupational status was related to lower depression level. By playing an inhibitory moderating role between the association of occupational status and depression, internet use and occupational status formed a substituting effect on impacting depression. Meanwhile, the above moderating effect was only significant in urban group and insignificant in rural group. Besides, being male and having higher social class were significant and negative predictors of depression.

ICT and society 5.0 promoted internet use among older adults, and this help alleviated health imbalance caused by unbalanced occupational status. For a society where the occupational status is closely linked to an individual's social resources, it is necessary to further promote internet penetration and health equalization.

I would like to express my deep appreciation for your response and the reviewers’ suggestions. I hope that the revised manuscript is now acceptable for publication. I look forward to hearing from you soon.

Sincerely yours,

Yujie Zhang

Shanghai Jiao Tong University

No. 1954 Hua Shan Road

Shanghai 200030, China

Tel. +86 18696960193

Email. zhangyujie@sjtu.edu.cn

Response to Reviewer 1 Comments

Point 1: The paper covers a promising topic, however, the realization is quite poor. The study is too short and the structure isn't eligible for a scientific writing.

Response 1: Thank you for your important suggestions, which were very helpful for the manuscript improvement.

I have carefully considered your suggestion and have made corresponding revisions.

Firstly, I expanded the manuscript to 5,442 words, so more emphasis could be given to literature review, empirical results presentation, and further discussion.

Secondly, I restructured the article by dividing it into five parts. 

Part 1 is Introduction. Paragraph 1 describes the importance to study the relationship among occupational status, internet use and depression. Paragraph 2 summarizes past studies about the relationship between occupational status and depression and put forward Hypothesis 1. Paragraph 3 highlights the emerging trend of information and communications technology (ICT) application and describes the concept of society 5.0, which explains the rationale of choosing internet use as a moderator. Paragraph 4 summarizes past studies of the relationship among occupational status, internet use and depression and put forward Hypothesis 2. Paragraph 5 highlights the importance of this study.

Part 2 is Methods. Data and sample, measures, statistical analysis and research framework are explained in detail.

Part 3 is Results. Descriptive statistics, hierarchical regression analyses, the moderating effect of internet use, analysis of heterogeneity of residence registration, and robustness test constituted this part.

Part 4 is Discussion. Paragraph 1 looks back the background of this study and point out the theoretical and practical contributions of this study. Paragraph 2 summarized findings to support Hypothesis 1 and compares this finding to similar results from past studies. Besides, paragraph 2 points out a possible chain reaction among occupational status, sex, and social class. Paragraph 3 summarized findings to support Hypothesis 2 and analyzes the rationales for this phenomenon. Paragraph 4 compares urban group with rural group and put forward further suggestions. Paragraph 5 discusses strength and limitations of this study. 

Part 5 is conclusion, which again point out the importance to pay attention to internet use in society 5.0 and promote health equalization among older adults.

Thank you again for your suggestions to better improve the article.

Point 2: Abstract isn't clear, title is too complicated and incomprehensive. Actually, the whole paper is incomprehensible.

Response 2: Thank you for your important suggestions, which were very helpful for the manuscript improvement.

I have carefully considered your suggestion and have made corresponding revisions.

Firstly, I revised the abstract. The abstract now is composed of four parts. Part 1 is Background, which describes the development of information and communications technology (ICT) and society 5.0 and the importance to study the relationship among occupational status, internet use and depression. Part 2 is Methods, which summarizes the data acquisition, data analysis and statistical framework. Part 3 is Results, which describes the main findings of this study and reports conclusions worth further attention. Part 4 is conclusion, which sheds light on policy implications for healthy aging in society 5.0.

Secondly, I changed the title. Now the title is as follows, namely, “The role of internet use in the relationship between occupational status and depression”. In this way, readers could more easily understand the research objects of this study. 

Thirdly, I edited the English writing for this article to make the article more smooth.

Thank you again for your suggestions to better improve the article.

Point 3: The literature review is totally missing, only very few sources were used for this study.

Response 2: Thank you for your important suggestions, which were very helpful for the manuscript improvement.

I have carefully considered your suggestion and have made corresponding revisions. By reorganizing the literature review, 35 sources were used in the literature review part. Following is the revised literature review.

In most societies, social members are often divided into different classes formally or informally due to their different characteristics such as power [1], property [2], education [3], family [4], race [5], gender [6], age [7], and occupation [8], that is, there is a system that gives different social members different social status. Occupational stratification is a very key feature to distinguish social members [9, 10]. In today's industrialized society, the level of occupational status often becomes an indicator of a person's status in society and affects people's job selection behavior and mobility trend [11]. Therefore, the change of occupational status and reputation has also become an important indicator to predict the direction and degree of social structure differentiation. Past studies have shown that social structure differentiation is related to depression [12, 13]. Entering an aging society, older population is a highly vulnerable group to depression [14]. It is very important to distinguish the mechanism of different occupational status on depression and the effect of ICT technical variable on this relationship.

Occupational status contributes to the acquisition of workers' socio-economic status and affects depression by affecting social resources accumulation, which has been confirmed by many empirical studies based on different societies [15,16]. Lower occupational status may relate to lower authority, salary, and social status, which increases vulnerability to anxiety and depression [17]. In addition, higher occupational status often provides individuals with more financial and social resources, which can promote good mental health [18]. Based on 10 years register data with 2,929 samples, a study found that people with higher occupational status would less likely troubled by mental disorders comparing with lower occupational status peers [19]. Although there was evidence that higher status occupation could relate to higher stress and may take up the time of family and friends and affect personal happiness [20], for older adults, the likelihood of negative impact caused by high occupational status is quite low. Because for people aged 60 and above, high occupational status more means work benefits and harmonious interpersonal relationships as the physical strength of older adults does not allow them to work long hours [21]. Especially in China, people generally retire before the age of 60. Therefore, high occupational status refers more to a kind of welfare [22]. Therefore, this study proposes the following hypothesis.

Hypothesis 1: Higher occupational status is significantly correlated with less depression. 

Notably, the application of information and communications technology (ICT) lay a good foundation for society 5.0. Society 5.0 is a concept highlighting the increasingly obvious impact of population aging on society, to meet the needs of economic and social development [23]. In society 5.0, ICT is widely used to realize an intelligent society with longevity for residents [24]. During COVID-19, prevention and control measures has accelerated the application of internet use to the daily life of older adults [25]. Especially in societies like China, where was at the intersection of the rapid development of internet technology and application and the rapid promotion of the process of population aging [26]. While expanding basic public services, promoting the better integration of older adults into society, and enhancing the sharing of information resources, the internet use is also facing new challenges and opportunities on how to better integrate into social governance and public service system and play a greater role in establishing a digitally inclusive friendly social environment for older adults. In 2020, the State Council of China issued the Implementation Plan on Effectively Solving the Difficulties in Using Intelligent Technology. It was proposed that by the end of 2022, the level of intelligent services for older adults should be significantly improved, online and offline services should be more collaborative, and long-term mechanism to solve the problem of digital divide for older adults should be basically established. This public policy also raises the question to further think: based on constantly bridging the digital divide, how can we make full use of the development of the Internet to bring opportunities for the implementation of the national strategy to actively respond to population aging and enrich the lives of older adults?

Past studies have suggested that internet use was positively related to occupational status. As policymakers always try to close the digital divide so that social mobility could be improved, a study based on the British Household Panel Survey (BHPS) and its successor, the UK Household Longitudinal Survey (UKHLS) found that internet use was a positive predictor of occupational related social status [27]. In turn, it was proven that occupational related social status would impact internet use. For example, some occupations required people to use internet for related information or connecting with co-workers [28]. For older adults, a study consisted of 17,704 older adults as samples found that low occupational related social status significantly decreased the possibility to use internet for getting health information, which may further decrease health status [29]. In addition to this indirect impact, internet use has been found to have significant impact on depression among older adults [30]. Older adults used internet for different purposes, such as communication with friends, information search, assignment execution and entertainment [31]. By using internet, older adults’ social capital could be significantly improved [32], which contributed to the alleviation of depression [33]. For retired older adults, this alleviation effect would become more prominent, as internet formed a virtual space for them to keep in touch with old friends and society, to gain social support [34]. After retirement, the continuous influence of occupational status will also be reflected in the network social capital [35]. In sum, internet use is closely connected with occupational status and depression, and possibly moderates the relationship between occupational status and depression. Therefore, this study proposes the following hypothesis.

Hypothesis 2: Internet use plays an inhibitory moderating role between occupational status and depression.

To the best of the author’s knowledge and based on this research, there were no previous studies concerning the relationship among occupational status, internet use and depression among older adults in China. This study was conducted to fulfill this research gap. The purpose of this study was to estimate the moderating role of internet use on the association between occupational status and depression in society 5.0.

Following is the newly added sources for literature review. To make the literature more updated, except for some important articles, most articles were published after 2018.

1. Rucker DD, Galinsky AD. Social power and social class: Conceptualization, consequences, and current challenges. Curr Opin Psychol. 2017; 18: 26-30.

2. Ware JK. Property value as a proxy of socioeconomic status in education. Educ Urban Soc. 2019; 51: 99-119.

3. Yan G, Peng Y, Hao Y, et al. Household head's educational level and household education expenditure in China: The mediating effect of social class identification. Int J Educ Dev. 2021; 83: 102400.

4. Pensiero N, Schoon I. Social inequalities in educational attainment: The changing impact of parents' social class, social status, education and family income, England 1986 and 2010. Longitud Life Course Stud. 2019; 10: 87-108.

5. Bumpus JP, Umeh Z, Harris AL. Social class and educational attainment: Do blacks benefit less from increases in parents’ social class status? Sociol Race Ethnic. 2020; 6: 223-241.

6. Dias FA. How skin color, class status, and gender intersect in the labor market: Evidence from a field experiment. Res Soc Stratif Mobil. 2020; 65: 100477.

7. Lahelma E, Pietiläinen O, Chandola T, et al. Occupational social class trajectories in physical functioning among employed women from midlife to retirement. BMC Public Health. 2019; 19: 1-10.

8. Kontogiannis N, Litina A, Varvarigos D. Occupation-induced status, social norms, and economic growth. J Econ Behav Organ. 2019; 163: 348-360.

9. Zhou X, Wodtke GT. Income stratification among occupational classes in the United States. Soc Forces. 2019; 97: 945-972.

10. Williams M, Bol T. Occupations and the wage structure: The role of occupational tasks in Britain. Res Soc Stratif Mobil. 2018; 53: 16-25.

11. Riederer B, Verwiebe R, Seewann L. Changing social stratification in Vienna: W hy are migrants declining from the middle of society? Popul Space Place. 2019; 25: e2215.

12. Bharat V, Habarth J, Keledjian N, et al. Association between subjective social status and facets of depression self‐stigma. J Community Psychol. 2020; 48: 1059-1065.

13. Sumner LA, Olmstead R, Azizoddin DR, et al. The contributions of socioeconomic status, perceived stress, and depression to disability in adults with systemic lupus erythematosus. Disabil Rehabil. 2020; 42: 1264-1269.

14. Barnett A, Zhang CJP, Johnston JM, et al. Relationships between the neighborhood environment and depression in older adults: A systematic review and meta-analysis. Int Psychogeriatr. 2018; 30: 1153-1176.

15. Domènech-Abella J, Mundó J, Leonardi M, et al. The association between socioeconomic status and depression among older adults in Finland, Poland and Spain: A comparative cross-sectional study of distinct measures and pathways. J Affect Disord. 2018; 241: 311-318.

16. Hori D, Takao S, Kawachi I, et al. Relationship between workplace social capital and suicidal ideation in the past year among employees in Japan: A cross-sectional study. BMC Public Health. 2019; 19: 1-11.

17. Gkiouleka A, Huijts T. Intersectional migration-related health inequalities in Europe: Exploring the role of migrant generation, occupational status & gender. Soc Sci Med. 2020; 267: 113218.

18. Freeland RE, Hoey J. The structure of deference: Modeling occupational status using affect control theory. Am Sociol Rev. 2018; 83: 243-277.

19. Castelpietra G, Balestrieri M, Bovenzi M. Occupational status and hospitalisation for mental disorders: Findings from Friuli Venezia Giulia region, Italy, 2008–2017. Int J Psychiat Clin. 2019; 23: 265-272.

20. Grace MK, VanHeuvelen JS. Occupational variation in burnout among medical staff: Evidence for the stress of higher status. Soc Sci Med. 2019; 232: 199-208.

21. Fernández-Niño JA, Bonilla-Tinoco LJ, Manrique-Espinoza BS, et al. Work status, retirement, and depression in older adults: An analysis of six countries based on the Study on Global Ageing and Adult Health (SAGE). SSM-Popul Health. 2018; 6: 1-8.

22. Feng J, Li Q, Smith J P. Retirement effect on health status and health behaviors in urban China. World Dev. 2020; 126: 104702.

23. Gladden ME. Who will be the members of Society 5.0? Towards an anthropology of technologically posthumanized future societies. Soc Sci. 2019; 8: 148.

24. 

24. Roblek V, Meško M, Bach MP, et al. The interaction between internet, sustainable development, and emergence of society 5.0. Data. 2020; 5: 80.

25. Seifert A. The digital exclusion of older adults during the COVID-19 pandemic. J Gerontol Soc Work. 2020; 63: 674-676.

26. Sun X, Yan W, Zhou H, et al. Internet use and need for digital health technology among the elderly: A cross-sectional survey in China. BMC Public Health. 2020; 20: 1-8.

27. Eynon R, Deetjen U, Malmberg LE. Moving on up in the information society? A longitudinal analysis of the relationship between Internet use and social class mobility in Britain. Inf Soc. 2018; 34: 316-327.

28. Kobul MK. Socioeconomic status influences Turkish digital natives’ internet use habitus. Behav Inf Technol. 2022: 1-19.

29. Yoon H, Jang Y, Vaughan PW, et al. Older adults’ internet use for health information: Digital divide by race/ethnicity and socioeconomic status. J Appl Gerontol. 2020; 39: 105-110.

30. Quintana D, Cervantes A, Sáez Y, et al. Internet use and psychological well-being at advanced age: Evidence from the English longitudinal study of aging. Int J Environ Res Public Health. 2018; 15: 480.

31. Lifshitz R, Nimrod G, Bachner YG. Internet use and well-being in later life: A functional approach. Aging Ment Health. 2018; 22: 85-91.

32. Barbosa Neves B, Fonseca JRS, Amaro F, et al. Social capital and Internet use in an age-comparative perspective with a focus on later life. PLoS One. 2018; 13: e0192119.

33. Han KM, Han C, Shin C, et al. Social capital, socioeconomic status, and depression in community-living elderly. J Psychiatr Res. 2018; 98: 133-140.

34. Macdonald B, Hülür G. Internet adoption in older adults: Findings from the Health and Retirement Study. Cyberpsychol. Behav. Soc. Netw. 2021; 24: 101-107.

35. Hunsaker A, Hargittai E. A review of Internet use among older adults. New Media Soc. 2018; 20: 3937-3954.

Thank you again for your suggestions to better improve the article.

Point 4: Methodology is not clear, selection reasons aren't explained.

Response 2: Thank you for your important suggestions, which were very helpful for the manuscript improvement.

I have carefully considered your suggestion and have made corresponding revisions. I revised the methodology part as follows.

In Data and Sample part, specific information about dataset and sample used in this study was explained in detail.

The cross-sectional data of this study was obtained from the 2017 wave of the Chinese General Social Survey (CGSS: http://cnsda.ruc.edu.cn/index.php?r=projects/view&id=94525591). The CGSS was conducted by the Survey and Data Center of Renmin University of China. At present, CGSS data has become the main data source for the study of Chinese society and is widely used in scientific research, teaching, and government decision-making. The stratified multiple-stage probability sampling method was used in the survey, which was carried out in all urban and rural households in 31 provinces, autonomous regions, and municipalities directly under the central government (excluding Hong Kong, Macao, and Taiwan), which systematically summarized the trend of social change. The sample size of this project was 12,582. The subjects of this study were older adults over 60 years old. After the screening and elimination of samples lacking relevant variables, 2,403 effective samples were obtained.

In Measures part, reasons for measure selection were explained in literature review, thus leaving Measures more focused on questionnaire description.

“Occupational status contributes to the acquisition of workers' socio-economic status and affects depression by affecting social resources accumulation, which has been confirmed by many empirical studies based on different societies. Lower occupational status may relate to lower authority, salary, and social status, which increases vulnerability to anxiety and depression. In addition, higher occupational status often provides individuals with more financial and social resources, which can promote good mental health. Based on 10 years register data with 2,929 samples, a study found that people with higher occupational status would less likely troubled by mental disorders comparing with lower occupational status peers. Although there was evidence that higher status occupation could relate to higher stress and may take up the time of family and friends and affect personal happiness, for older adults, the likelihood of negative impact caused by high occupational status is quite low. Because for people aged 60 and above, high occupational status more means work benefits and harmonious interpersonal relationships as the physical strength of older adults does not allow them to work long hours. Especially in China, people generally retire before the age of 60. Therefore, high occupational status refers more to a kind of welfare.”

“Past studies have suggested that internet use was positively related to occupational status. As policymakers always try to close the digital divide so that social mobility could be improved, a study based on the British Household Panel Survey (BHPS) and its successor, the UK Household Longitudinal Survey (UKHLS) found that internet use was a positive predictor of occupational related social status. In turn, it was proven that occupational related social status would impact internet use. For example, some occupations required people to use internet for related information or connecting with co-workers. For older adults, a study consisted of 17,704 older adults as samples found that low occupational related social status significantly decreased the possibility to use internet for getting health information, which may further decrease health status. In addition to this indirect impact, internet use has been found to have significant impact on depression among older adults [30]. Older adults used internet for different purposes, such as communication with friends, information search, assignment execution and entertainment. By using internet, older adults’ social capital could be significantly improved, which contributed to the alleviation of depression. For retired older adults, this alleviation effect would become more prominent, as internet formed a virtual space for them to keep in touch with old friends and society, to gain social support. After retirement, the continuous influence of occupational status will also be reflected in the network social capital. In sum, internet use is closely connected with occupational status and depression, and possibly moderates the relationship between occupational status and depression.”

In Statistical Analysis part, all statistical methods and steps were explained in detail and a research framework figure was added to present more vividly.

Figure 1. Research framework.

Thank you again for your suggestions to better improve the article.

Point 4: Based on the results, the evaluation is not possible.The role of internet use is not explained. 

Response 2: Thank you for your important suggestions, which were very helpful for the manuscript improvement.

I have carefully considered your suggestion and have made corresponding revisions. In the Results part, I put more emphasis on explaining the role of internet.

Firstly, by constructing hierarchical regression analysis, it was suggested that internet use may play an inhibitory moderating role between occupational status and depression and further investigation should be conducted. In the third step, occupational status was negatively correlated with depression (coefficient = -0.126, P < 0.001). Internet use was negatively correlated with depression, too (coefficient = -0.132, P < 0.01). Notably, the interaction item of occupational status and internet use was positively correlated with depression (coefficient = 0.026, P < 0.05).

Table 2. Hierarchical regression results (N=2,403).

Notes: t values in parentheses, ***p<0.001, **p<0.01, *p<0.05. 

Secondly, by implementing moderating effect analysis, the inhibitory moderating effect of internet use in the relationship between occupational status and depression was presented.

Table 3 illustrates the moderating effect of internet use. After centralization of occupational status and internet use, the regression results revealed that internet use played an inhibitory role in moderating the association between occupational status and depression (coefficient = 0.026, P < 0.05). Moreover, sex, social class and residence registration were significantly negatively related to depression. Besides, the marginal effect of occupational status and internet use was significant and negative, which suggests that there may exit substituting effect between the two variables.

Table 3. Moderating effect analysis (N=2,403).

Notes: ***p<0.001, **p<0.01, *p<0.05. 

Figure 2 shows the inhibitory moderating effect of internet use in the relationship between occupational status and depression. This means that when older adults had higher level of internet use, the same occupational status score increase would lead to less depression reduction. Although occupational status had a significant negative effect on depression, there were the following conditions: the negative role of occupational status was relatively strong when internet use was low, but the negative role of occupational status gradually decreased with an increase in internet use. In sum, the results supported Hypothesis 2.

Figure 2. Moderating effect of internet use.

Thank you again for your suggestions to better improve the article.

Point 5: I recommend to rewrite it by quality standards and resubmit it again.

Response 2: Thank you for your important suggestions, which were very helpful for the manuscript improvement.

I have carefully considered your suggestion and have rewritten the manuscript by quality standards. Moreover, a heterogeneity analysis based on residence registration was added, so the reader could further understand of the relationship among occupational status, internet use and depression between urban and rural areas. The analysis of heterogeneity of residence registration is as follows.

Table 4 shows the results for heterogeneity analysis. For urban group, the moderating effect of internet use was significant, and internet use had an inhibitory effect on the association between occupational status and depression. Meanwhile, control variables such as sex and social class were significant and negative predictors of depression in the urban group. For rural group, the moderating effect of internet use was insignificant. Except for education and social class, other control variables were not significant in explaining depression in the rural group.

Table 4. Residence registration heterogeneity analysis (N=2,403).

Notes: t values in parentheses, ***p<0.001, **p<0.01, *p<0.05. 

Moreover, the results suggest that only in urban group, the moderating effect of internet use was significant whereas in rural group, the moderating effect of internet use was insignificant. The reason for this contradiction could be the difference in internet penetration rate between urban and rural areas in China. Due to urbanization, most young people left their rural hometowns to work in urban areas, thus there was a lack of young people who could teach older adults to use the Internet. Besides, the rural economy was not as developed as the cities, so not everyone could afford the cost of using the Internet. At present, the Chinese government is vigorously advocating Rural Revitalization Plan. In the future, policies should be more inclined to rural areas and popularize Internet use in rural areas

Thank you again for your suggestions to better improve the article.

Response to Reviewer 2 Comments

Point 1: The paper presents the results of a methodologically sound, logically conducted study. The objectives and results are clearly defined by the authors and the modelling of the research process is professional. The results and main conclusions are formulated. This study not only investigated the effects of work unit’s ownership form on the mental health of the elderly, but also introduced internet use as a moderating variable to analyze the impact mechanism. With the acceleration of the digital process, this study once again proves that internet can bridge the mental health gap caused by the nature of the work unit. Main conclusions are: Work unit’s ownership form could significantly impact the mental health of the elderly. Internet use inhibits the relationship between work unit’s ownership form and mental health promotion. Their findings suggest that work unit’s ownership form was significantly associated with mental health status for older Chinese residents, moderated by internet use.

While it is true that the study focuses on the specificity of China, I think that it will be worthwhile to look outside and make comparisons with other countries as a continuation of the research even if Chinese work units have unique attributes.

Response 1: Thank you for your useful suggestions, which were very helpful for the improvement of the manuscript.

To make comparisons with other countries could enrich this study with comparison values. Thus, I made the corresponding adjustments.

Firstly, in the Literature Review part. More studies about the relationship among occupational status, internet use and depression were added as follows.

“Occupational status contributes to the acquisition of workers' socio-economic status and affects depression by affecting social resources accumulation, which has been confirmed by many empirical studies based on different societies. Lower occupational status may relate to lower authority, salary, and social status, which increases vulnerability to anxiety and depression. In addition, higher occupational status often provides individuals with more financial and social resources, which can promote good mental health. Based on 10 years register data with 2,929 samples, a study found that people with higher occupational status would less likely troubled by mental disorders comparing with lower occupational status peers. Although there was evidence that higher status occupation could relate to higher stress and may take up the time of family and friends and affect personal happiness, for older adults, the likelihood of negative impact caused by high occupational status is quite low. Because for people aged 60 and above, high occupational status more means work benefits and harmonious interpersonal relationships as the physical strength of older adults does not allow them to work long hours. Especially in China, people generally retire before the age of 60. Therefore, high occupational status refers more to a kind of welfare.”

“Past studies have suggested that internet use was positively related to occupational status. As policymakers always try to close the digital divide so that social mobility could be improved, a study based on the British Household Panel Survey (BHPS) and its successor, the UK Household Longitudinal Survey (UKHLS) found that internet use was a positive predictor of occupational related social status. In turn, it was proven that occupational related social status would impact internet use. For example, some occupations required people to use internet for related information or connecting with co-workers. For older adults, a study consisted of 17,704 older adults as samples found that low occupational related social status significantly decreased the possibility to use internet for getting health information, which may further decrease health status. In addition to this indirect impact, internet use has been found to have significant impact on depression among older adults [30]. Older adults used internet for different purposes, such as communication with friends, information search, assignment execution and entertainment. By using internet, older adults’ social capital could be significantly improved, which contributed to the alleviation of depression. For retired older adults, this alleviation effect would become more prominent, as internet formed a virtual space for them to keep in touch with old friends and society, to gain social support. After retirement, the continuous influence of occupational status will also be reflected in the network social capital. In sum, internet use is closely connected with occupational status and depression, and possibly moderates the relationship between occupational status and depression.”

Secondly, in the Conclusion part, further policy implications for societies like China were made as follows. 

“The findings emphasize that internet use is an important moderator in the relationship between occupational status and depression. The results may be useful for the implementation of active ageing strategies. Society 5.0 may be difficult to realize, especially in low- and middle-income countries, where ICT investment resources are scarce; thus, ICT application policies may seem difficult to implement. It is much better to utilize existing internet infrastructure, as this choice is more practical. Last, the findings of this study suggest that further attention should paid to promote internet use habits to the comprehensive improvement of older adults’ health abilities to form a healthy aging society.”

Thirdly, I pointed out future studies should pay more attention to comparative studies as it worth investigating whether there exist differences in the attributes of occupational status.

Thank you again for your suggestions to improve the article.

Point 2: The link between the descriptive part of the study and the empirical phase would have been further strengthened by supporting the research background and the precise research objectives with literature and preliminary research results.

Response 2: Thank you for pointing this out, which was very helpful for improving the article.

Firstly, to present more precise research objectives, I used “depression” to substitute “mental health” and “occupational status” to replace “work unit’s ownership form” to more accurately reflect the dependent variable and independent variable studied. 

Secondly, to support the research background, I added more information in the Introduction part as follows.

“In most societies, social members are often divided into different classes formally or informally due to their different characteristics such as power, property, education, family, race, gender, age, and occupation, that is, there is a system that gives different social members different social status. Occupational stratification is a very key feature to distinguish social members. In today's industrialized society, the level of occupational status often becomes an indicator of a person's status in society and affects people's job selection behavior and mobility trend. Therefore, the change of occupational status and reputation has also become an important indicator to predict the direction and degree of social structure differentiation. Past studies have shown that social structure differentiation is related to depression. Entering an aging society, older population is a highly vulnerable group to depression. It is very important to distinguish the mechanism of different occupational status on depression and the effect of ICT technical variable on this relationship.”

Thirdly, to better link the descriptive part of the study and the empirical phase, more literature and preliminary research results were presented both in the Literature Review part and Discussion part.

In the Literature Review part, preliminary research results were presented to support Hypothesis 1 and Hypothesis 2.

“Occupational status contributes to the acquisition of workers' socio-economic status and affects depression by affecting social resources accumulation, which has been confirmed by many empirical studies based on different societies [15,16]. Lower occupational status may relate to lower authority, salary, and social status, which increases vulnerability to anxiety and depression [17]. In addition, higher occupational status often provides individuals with more financial and social resources, which can promote good mental health [18]. Based on 10 years register data with 2,929 samples, a study found that people with higher occupational status would less likely troubled by mental disorders comparing with lower occupational status peers [19]. Although there was evidence that higher status occupation could relate to higher stress and may take up the time of family and friends and affect personal happiness [20], for older adults, the likelihood of negative impact caused by high occupational status is quite low. Because for people aged 60 and above, high occupational status more means work benefits and harmonious interpersonal relationships as the physical strength of older adults does not allow them to work long hours [21]. Especially in China, people generally retire before the age of 60. Therefore, high occupational status refers more to a kind of welfare [22]. Therefore, this study proposes the following hypothesis.

Hypothesis 1: Higher occupational status is significantly correlated with less depression. 

Notably, the application of information and communications technology (ICT) lay a good foundation for society 5.0. Society 5.0 is a concept highlighting the increasingly obvious impact of population aging on society, to meet the needs of economic and social development [23]. In society 5.0, ICT is widely used to realize an intelligent society with longevity for residents [24]. During COVID-19, prevention and control measures has accelerated the application of internet use to the daily life of older adults [25]. Especially in societies like China, where was at the intersection of the rapid development of internet technology and application and the rapid promotion of the process of population aging [26]. While expanding basic public services, promoting the better integration of older adults into society, and enhancing the sharing of information resources, the internet use is also facing new challenges and opportunities on how to better integrate into social governance and public service system and play a greater role in establishing a digitally inclusive friendly social environment for older adults. In 2020, the State Council of China issued the Implementation Plan on Effectively Solving the Difficulties in Using Intelligent Technology. It was proposed that by the end of 2022, the level of intelligent services for older adults should be significantly improved, online and offline services should be more collaborative, and long-term mechanism to solve the problem of digital divide for older adults should be basically established. This public policy also raises the question to further think: based on constantly bridging the digital divide, how can we make full use of the development of the Internet to bring opportunities for the implementation of the national strategy to actively respond to population aging and enrich the lives of older adults?

Past studies have suggested that internet use was positively related to occupational status. As policymakers always try to close the digital divide so that social mobility could be improved, a study based on the British Household Panel Survey (BHPS) and its successor, the UK Household Longitudinal Survey (UKHLS) found that internet use was a positive predictor of occupational related social status [27]. In turn, it was proven that occupational related social status would impact internet use. For example, some occupations required people to use internet for related information or connecting with co-workers [28]. For older adults, a study consisted of 17,704 older adults as samples found that low occupational related social status significantly decreased the possibility to use internet for getting health information, which may further decrease health status [29]. In addition to this indirect impact, internet use has been found to have significant impact on depression among older adults [30]. Older adults used internet for different purposes, such as communication with friends, information search, assignment execution and entertainment [31]. By using internet, older adults’ social capital could be significantly improved [32], which contributed to the alleviation of depression [33]. For retired older adults, this alleviation effect would become more prominent, as internet formed a virtual space for them to keep in touch with old friends and society, to gain social support [34]. After retirement, the continuous influence of occupational status will also be reflected in the network social capital [35]. In sum, internet use is closely connected with occupational status and depression, and possibly moderates the relationship between occupational status and depression. Therefore, this study proposes the following hypothesis.

Hypothesis 2: Internet use plays an inhibitory moderating role between occupational status and depression.”

In the Discussion part, preliminary research results were presented to explain the findings and the underlying rationales.

“As COVID-19 accelerated the application of ICT and construction process of society 5.0 [43], the importance of the role internet use plays between occupational status and depression should be paid further attention. For older adults, many of they have entered retirement, and the impact of occupational status on their lives has basically solidified, including social connections [44], social capital [45], and economic and social resources [46]. Therefore, the impact on depression is difficult to change. However, bridging the digital divide may reduce the health inequality caused by occupational status inequality and promote the equalization of mental health of older adults. This cross-sectional study was designed to assess the moderating role of internet use between occupational status and depression to provide theoretical and policy implications for healthy aging.

The current study showed that most older adults had medium level of depression. And the empirical findings suggest that a higher occupational status was correlated with lower depression, which confirmed findings from previous studies. A study of 1,382 samples during COVID-19 in Vietnam found that lockdown policies had changed people’s occupational status, which led to the change in the level of stress and depression [47]. Notably, in a sample consisted of 23,247 older adults aged between 65 and 88 testified the hypothesis that higher occupational status was positively related to less depressive symptoms [48]. In this study, being male and having higher social status were also correlated with less depression. It is possible that male adults were more likely to gain higher occupational status, thus having higher social status. More attention should be paid to this chain reaction as people of different genders will be treated differently in their careers [49].

When internet use was added as a moderator, it was found that internet use formed a substitution effect with occupational status on impacting depression, which caused an inhibitory moderating effect. One contributing factor to this phenomenon could be accumulated social capital caused by internet use [50]. Though previous occupational status affected depression, as older adults more and more used the Internet for friends’ interactions and entertainment, they could establish new connections with society and not constrained by previous occupational status. Another contributing factor could be health information acquisition caused by internet use [51]. Holding other conditions constant, health knowledge acquired from the Internet could benefit older adults in gaining professional help to alleviate depression. For example, some older adults may rely on internet apps for psychological counseling [52].

Moreover, the results suggest that only in urban group, the moderating effect of internet use was significant whereas in rural group, the moderating effect of internet use was insignificant. The reason for this contradiction could be the difference in internet penetration rate between urban and rural areas in China [53]. Due to urbanization, most young people left their rural hometowns to work in urban areas, thus there was a lack of young people who could teach older adults to use the Internet [54]. Besides, the rural economy was not as developed as the cities, so not everyone could afford the cost of using the Internet [55]. At present, the Chinese government is vigorously advocating Rural Revitalization Plan. In the future, policies should be more inclined to rural areas and popularize Internet use in rural areas”

Thank you again for your suggestions to improve the article.

Point 3: The study aims to explore how the nature of work units affects the mental health of the elderly, and to explore the regulatory role of Internet applications in the era of digital governance. The authors have met this objective within the scope of the study. 

The limitations of the study are mentioned, and plans to overcome them should be shared with the reader in more detail, so that they can see more clearly the research model the authors intend to follow in pursuing data collection.

Response 3: Thank you for your important suggestion, which was very helpful for improving the article.

The plans to overcome the limitation were added in the Strength and Limitations part as follows.

“A limitation is that the results were cross-sectional; thus, they only examined the relationships among occupational status, internet use, and depression. Longitudinal and experimental studies should be conducted to reveal the causal relationship among the above variables.”

“Meanwhile, as this study mainly focused on Chinese occupational status, comparative studies should be paid more attention to investigate whether there exist differences in the attributes of occupation status in cross-cultural context.”

In this way, readers could anticipate that in the future, the author may conduct longitudinal empirical study or experiment study to overcome cross-sectional design’s limitation and may conduct comparative studies to understand the nature of occupational status in different countries.

Thank you again for your suggestions to improve the article.

Point 4: I consider the strength of the study to be its methodology and its very thorough and professional presentation. The reader is presented with a logical research process model and a presentation of results.

Even if the results are regional, I think that they can serve as a good example for further researchers and for further studies on similar topics. The results presented in this study have many relevant elements not only from a theoretical point of view but also from a practical point of view.

Response 4: Thank you for your affirmation of my research.

To make the methodology and presentation more thorough, I added two parts in the Results. 

Part 1 is Hierarchical regression analyses, so the readers can have a better idea of how occupational status, internet use and depression interacted with each other.

Table 2 shows the results of hierarchical regression analyses. In the first step, occupational status was negatively correlated with depression (coefficient = -0.078, P < 0.001). Control variables including sex, education, social class, and residence registration significantly explained depression in a negative way. In the second step, occupational status was negatively correlated with depression (coefficient = -0.074, P < 0.001). Control variables including sex, social class and residence registration significantly explained depression in a negative way. In the third step, occupational status was negatively correlated with depression (coefficient = -0.126, P < 0.001). Internet use was negatively correlated with depression, too (coefficient = -0.132, P < 0.01). Notably, the interaction item of occupational status and internet use was positively correlated with depression (coefficient = 0.026, P < 0.05). Control variables explained depression in the same way with which in Step 2. It was suggested that internet use may play an inhibitory moderating role between occupational status and depression and further investigation should be conducted. In sum, the results supported Hypothesis 1.

Table 2. Hierarchical regression results (N=2,403).

Notes: t values in parentheses, ***p<0.001, **p<0.01, *p<0.05. 

Part 2 is Analysis of heterogeneity of residence registration, so the readers could have a better idea of how the differences between rural and urban areas in China shaped the relationship among occupational status, internet use and depression.

Table 5 shows the results for heterogeneity analysis. For urban group, the moderating effect of internet use was significant, and internet use had an inhibitory effect on the association between occupational status and depression. Meanwhile, control variables such as sex and social class were significant and negative predictors of depression in the urban group. For rural group, the moderating effect of internet use was insignificant. Except for education and social class, other control variables were not significant in explaining depression in the rural group.

Table 5. Residence registration heterogeneity analysis (N=2,403).

Notes: t values in parentheses, ***p<0.001, **p<0.01, *p<0.05. 

Moreover, after adding these two parts in the Results, more discussions about theoretical and practical implications were added. When internet use was added as a moderator, it was found that internet use formed a substitution effect with occupational status on impacting depression, which caused an inhibitory moderating effect. One contributing factor to this phenomenon could be accumulated social capital caused by internet use. Though previous occupational status affected depression, as older adults more and more used the Internet for friends’ interactions and entertainment, they could establish new connections with society and not constrained by previous occupational status. Another contributing factor could be health information acquisition caused by internet use. Holding other conditions constant, health knowledge acquired from the Internet could benefit older adults in gaining professional help to alleviate depression. For example, some older adults may rely on internet apps for psychological counseling. Moreover, the results suggest that only in urban group, the moderating effect of internet use was significant whereas in rural group, the moderating effect of internet use was insignificant. The reason for this contradiction could be the difference in internet penetration rate between urban and rural areas in China. Due to urbanization, most young people left their rural hometowns to work in urban areas, thus there was a lack of young people who could teach older adults to use the Internet. Besides, the rural economy was not as developed as the cities, so not everyone could afford the cost of using the Internet. At present, the Chinese government is vigorously advocating Rural Revitalization Plan. In the future, policies should be more inclined to rural areas and popularize Internet use in rural areas

Thank you again for your affirmation.

Response to Reviewer 3 Comments

Point 1: The topic of the paper is really interesting and in the face of the emerging novel ICT and society 5.0 extremely relevant as well. 

While the importance and effect of work units was emphasised and underlined with references, the internet usage and its potential relation with mental health was not addressed directly in the introductory/literature review part. The first mention of related literature can be found in the discussion part.

Response 1: Thank you for your useful suggestions, which were very helpful for the improvement of the manuscript.

Firstly, in order to more precisely reflect the independent and dependent variables, I renamed “work units’ ownership form” as “occupational status” and renamed “mental health” as “depression”. Secondly, I added literature to support the relationship among occupational status, internet use and depression both in the Introduction and Discussion part as follows.

In Introduction part.

“In most societies, social members are often divided into different classes formally or informally due to their different characteristics such as power [1], property [2], education [3], family [4], race [5], gender [6], age [7], and occupation [8], that is, there is a system that gives different social members different social status. Occupational stratification is a very key feature to distinguish social members [9, 10]. In today's industrialized society, the level of occupational status often becomes an indicator of a person's status in society and affects people's job selection behavior and mobility trend [11]. Therefore, the change of occupational status and reputation has also become an important indicator to predict the direction and degree of social structure differentiation. Past studies have shown that social structure differentiation is related to depression [12, 13]. Entering an aging society, older population is a highly vulnerable group to depression [14]. It is very important to distinguish the mechanism of different occupational status on depression and the effect of ICT technical variable on this relationship.

Occupational status contributes to the acquisition of workers' socio-economic status and affects depression by affecting social resources accumulation, which has been confirmed by many empirical studies based on different societies [15,16]. Lower occupational status may relate to lower authority, salary, and social status, which increases vulnerability to anxiety and depression [17]. In addition, higher occupational status often provides individuals with more financial and social resources, which can promote good mental health [18]. Based on 10 years register data with 2,929 samples, a study found that people with higher occupational status would less likely troubled by mental disorders comparing with lower occupational status peers [19]. Although there was evidence that higher status occupation could relate to higher stress and may take up the time of family and friends and affect personal happiness [20], for older adults, the likelihood of negative impact caused by high occupational status is quite low. Because for people aged 60 and above, high occupational status more means work benefits and harmonious interpersonal relationships as the physical strength of older adults does not allow them to work long hours [21]. Especially in China, people generally retire before the age of 60. Therefore, high occupational status refers more to a kind of welfare [22]. Therefore, this study proposes the following hypothesis.

Hypothesis 1: Higher occupational status is significantly correlated with less depression. 

Notably, the application of information and communications technology (ICT) lay a good foundation for society 5.0. Society 5.0 is a concept highlighting the increasingly obvious impact of population aging on society, to meet the needs of economic and social development [23]. In society 5.0, ICT is widely used to realize an intelligent society with longevity for residents [24]. During COVID-19, prevention and control measures has accelerated the application of internet use to the daily life of older adults [25]. Especially in societies like China, where was at the intersection of the rapid development of internet technology and application and the rapid promotion of the process of population aging [26]. While expanding basic public services, promoting the better integration of older adults into society, and enhancing the sharing of information resources, the internet use is also facing new challenges and opportunities on how to better integrate into social governance and public service system and play a greater role in establishing a digitally inclusive friendly social environment for older adults. In 2020, the State Council of China issued the Implementation Plan on Effectively Solving the Difficulties in Using Intelligent Technology. It was proposed that by the end of 2022, the level of intelligent services for older adults should be significantly improved, online and offline services should be more collaborative, and long-term mechanism to solve the problem of digital divide for older adults should be basically established. This public policy also raises the question to further think: based on constantly bridging the digital divide, how can we make full use of the development of the Internet to bring opportunities for the implementation of the national strategy to actively respond to population aging and enrich the lives of older adults?

Past studies have suggested that internet use was positively related to occupational status. As policymakers always try to close the digital divide so that social mobility could be improved, a study based on the British Household Panel Survey (BHPS) and its successor, the UK Household Longitudinal Survey (UKHLS) found that internet use was a positive predictor of occupational related social status [27]. In turn, it was proven that occupational related social status would impact internet use. For example, some occupations required people to use internet for related information or connecting with co-workers [28]. For older adults, a study consisted of 17,704 older adults as samples found that low occupational related social status significantly decreased the possibility to use internet for getting health information, which may further decrease health status [29]. In addition to this indirect impact, internet use has been found to have significant impact on depression among older adults [30]. Older adults used internet for different purposes, such as communication with friends, information search, assignment execution and entertainment [31]. By using internet, older adults’ social capital could be significantly improved [32], which contributed to the alleviation of depression [33]. For retired older adults, this alleviation effect would become more prominent, as internet formed a virtual space for them to keep in touch with old friends and society, to gain social support [34]. After retirement, the continuous influence of occupational status will also be reflected in the network social capital [35]. In sum, internet use is closely connected with occupational status and depression, and possibly moderates the relationship between occupational status and depression. Therefore, this study proposes the following hypothesis.

Hypothesis 2: Internet use plays an inhibitory moderating role between occupational status and depression.”

In Discussion part.

“When internet use was added as a moderator, it was found that internet use formed a substitution effect with occupational status on impacting depression, which caused an inhibitory moderating effect. One contributing factor to this phenomenon could be accumulated social capital caused by internet use [50]. Though previous occupational status affected depression, as older adults more and more used the Internet for friends’ interactions and entertainment, they could establish new connections with society and not constrained by previous occupational status. Another contributing factor could be health information acquisition caused by internet use [51]. Holding other conditions constant, health knowledge acquired from the Internet could benefit older adults in gaining professional help to alleviate depression. For example, some older adults may rely on internet apps for psychological counseling [52].

Moreover, the results suggest that only in urban group, the moderating effect of internet use was significant whereas in rural group, the moderating effect of internet use was insignificant. The reason for this contradiction could be the difference in internet penetration rate between urban and rural areas in China [53]. Due to urbanization, most young people left their rural hometowns to work in urban areas, thus there was a lack of young people who could teach older adults to use the Internet [54]. Besides, the rural economy was not as developed as the cities, so not everyone could afford the cost of using the Internet [55]. At present, the Chinese government is vigorously advocating Rural Revitalization Plan. In the future, policies should be more inclined to rural areas and popularize Internet use in rural areas”

Thank you again for your suggestions to improve the article.

Point 2: What is more, the relation of work unit ownership form and internet use is not explored, even though they are suggested to have strong correlation according to CFPS.

Response 2: Thank you for pointing this out, which was very helpful for improving the article.

In Results, I added Hierarchical Regression Analyses and The Moderating Effect of Internet Use to explore the relation of occupational status and internet use. It was found that occupational status and internet use formed a substituting effect on impacting depression.

From Table 2, it could be seen that internet use may play an inhibitory moderating role between occupational status and depression and further investigation should be conducted.

Table 2. Hierarchical regression results (N=2,403).

Notes: t values in parentheses, ***p<0.001, **p<0.01, *p<0.05. 

From Table 3, it could be seen that the marginal effect of occupational status and internet use was significant and negative, which suggests that there may exit substituting effect between the two variables.

Table 3. Moderating effect analysis (N=2,403)

Notes: ***p<0.001, **p<0.01, *p<0.05. 

Moreover, thanks to your valuable advice, I empirically tested the relationship between occupational status and internet use. It was found that in CGSS, occupational status and internet use had significant positive correlation, so I discussed the relation both in the empirical and discussion part to shed more light on policy implications. 

In Results. 

“Table 4 shows the correlation between occupational status and internet use. This means that higher internet use was significantly positively correlated with higher occupational status (coefficient = 0.036, P < 0.01). Age and residence registration were two positive predictors of occupational status whereas sex, education, income and social class were insignificant in predicting occupational status.”

Table 4. Correlation between occupational status and internet use (N=2,403).

Notes: ***p<0.001, **p<0.01, *p<0.05. 

In Discussion.

“In addition, internet use was significantly positively related to occupational status, thus, promoting internet use could be a mean to increase occupational status for those older adults who were still working, which would in turn decrease their depression.”

Thank you again for your suggestions to improve the article.

Point 3: The chosen methodology is in line with the research question and the aim and purpose of the research, however, the data presented in the paper could be enriched along the previous logic.

Response 3: Thank you for your important suggestion, which was very helpful for improving the article.

To enrich the data presented in the paper along the previous logic, I added previous studies both in the Introduction and Discussion part.

In the Introduction part.

For occupational status and depression.

“Occupational status contributes to the acquisition of workers' socio-economic status and affects depression by affecting social resources accumulation, which has been confirmed by many empirical studies based on different societies [15,16]. Lower occupational status may relate to lower authority, salary, and social status, which increases vulnerability to anxiety and depression [17]. In addition, higher occupational status often provides individuals with more financial and social resources, which can promote good mental health [18]. Based on 10 years register data with 2,929 samples, a study found that people with higher occupational status would less likely troubled by mental disorders comparing with lower occupational status peers [19]. Although there was evidence that higher status occupation could relate to higher stress and may take up the time of family and friends and affect personal happiness [20], for older adults, the likelihood of negative impact caused by high occupational status is quite low. Because for people aged 60 and above, high occupational status more means work benefits and harmonious interpersonal relationships as the physical strength of older adults does not allow them to work long hours [21]. Especially in China, people generally retire before the age of 60. Therefore, high occupational status refers more to a kind of welfare [22].”

For the moderating role of internet use between occupational status and depression.

“Past studies have suggested that internet use was positively related to occupational status. As policymakers always try to close the digital divide so that social mobility could be improved, a study based on the British Household Panel Survey (BHPS) and its successor, the UK Household Longitudinal Survey (UKHLS) found that internet use was a positive predictor of occupational related social status [27]. In turn, it was proven that occupational related social status would impact internet use. For example, some occupations required people to use internet for related information or connecting with co-workers [28]. For older adults, a study consisted of 17,704 older adults as samples found that low occupational related social status significantly decreased the possibility to use internet for getting health information, which may further decrease health status [29]. In addition to this indirect impact, internet use has been found to have significant impact on depression among older adults [30]. Older adults used internet for different purposes, such as communication with friends, information search, assignment execution and entertainment [31]. By using internet, older adults’ social capital could be significantly improved [32], which contributed to the alleviation of depression [33]. For retired older adults, this alleviation effect would become more prominent, as internet formed a virtual space for them to keep in touch with old friends and society, to gain social support [34]. After retirement, the continuous influence of occupational status will also be reflected in the network social capital [35]. In sum, internet use is closely connected with occupational status and depression, and possibly moderates the relationship between occupational status and depression.”

In the Discussion part.

For occupational status and depression.

“The current study showed that most older adults had medium level of depression. And the empirical findings suggest that a higher occupational status was correlated with lower depression, which confirmed findings from previous studies. A study of 1,382 samples during COVID-19 in Vietnam found that lockdown policies had changed people’s occupational status, which led to the change in the level of stress and depression [47]. Notably, in a sample consisted of 23,247 older adults aged between 65 and 88 testified the hypothesis that higher occupational status was positively related to less depressive symptoms [48]. In this study, being male and having higher social status were also correlated with less depression. It is possible that male adults were more likely to gain higher occupational status, thus having higher social status. More attention should be paid to this chain reaction as people of different genders will be treated differently in their careers [49].”

For the moderating role of internet use between occupational status and depression.

“When internet use was added as a moderator, it was found that internet use formed a substitution effect with occupational status on impacting depression, which caused an inhibitory moderating effect. One contributing factor to this phenomenon could be accumulated social capital caused by internet use [50]. Though previous occupational status affected depression, as older adults more and more used the Internet for friends’ interactions and entertainment, they could establish new connections with society and not constrained by previous occupational status. Another contributing factor could be health information acquisition caused by internet use [51]. Holding other conditions constant, health knowledge acquired from the Internet could benefit older adults in gaining professional help to alleviate depression. For example, some older adults may rely on internet apps for psychological counseling [52]. In addition, internet use was significantly positively related to occupational status, thus, promoting internet use could be a mean to increase occupational status for those older adults who were still working, which would in turn decrease their depression.”

Thank you again for your suggestions to improve the article.

Point 4: While author references 30 relevant international sources, only 3 from past 5 years, no form past 2, even though Covid19 has reshaped the economic and societal landscape of China as well. Author is recommend to read up on and refer to novel national and interantional literature especially realted to internet usage in different social and age groups.

Response 4: Thank you for pointing out this direction for the improvement of my manuscript.

Firstly, relevant international sources have been updated from 30 to 55 and most of them were from past 5 years, 1/3 of them were from past 2 years.

Secondly, considering COVID-19 has reshaped the economic and societal landscape of China, literature related to internet use in different social and age groups were added. Moreover, studies focused on the impact of COVID-19 have been added as well. For example:

24. Roblek V, Meško M, Bach MP, et al. The interaction between internet, sustainable development, and emergence of society 5.0. Data. 2020; 5: 80.

25. Seifert A. The digital exclusion of older adults during the COVID-19 pandemic. J Gerontol Soc Work. 2020; 63: 674-676.

26. Sun X, Yan W, Zhou H, et al. Internet use and need for digital health technology among the elderly: A cross-sectional survey in China. BMC Public Health. 2020; 20: 1-8.

27. Eynon R, Deetjen U, Malmberg LE. Moving on up in the information society? A longitudinal analysis of the relationship between Internet use and social class mobility in Britain. Inf Soc. 2018; 34: 316-327.

28. Kobul MK. Socioeconomic status influences Turkish digital natives’ internet use habitus. Behav Inf Technol. 2022: 1-19.

29. Yoon H, Jang Y, Vaughan PW, et al. Older adults’ internet use for health information: Digital divide by race/ethnicity and socioeconomic status. J Appl Gerontol. 2020; 39: 105-110.

30. Quintana D, Cervantes A, Sáez Y, et al. Internet use and psychological well-being at advanced age: Evidence from the English longitudinal study of aging. Int J Environ Res Public Health. 2018; 15: 480.

31. Lifshitz R, Nimrod G, Bachner YG. Internet use and well-being in later life: A functional approach. Aging Ment Health. 2018; 22: 85-91.

32. Barbosa Neves B, Fonseca JRS, Amaro F, et al. Social capital and Internet use in an age-comparative perspective with a focus on later life. PLoS One. 2018; 13: e0192119.

34. Macdonald B, Hülür G. Internet adoption in older adults: Findings from the Health and Retirement Study. Cyberpsychol. Behav. Soc. Netw. 2021; 24: 101-107.

35. Hunsaker A, Hargittai E. A review of Internet use among older adults. New Media Soc. 2018; 20: 3937-3954.

37. Szabo A, Allen J, Stephens C, et al. Longitudinal analysis of the relationship between purposes of internet use and well-being among older adults. Gerontologist. 2019; 59: 58-68.

43. Islam A, Islam M, Hossain Uzir MU, et al. The panorama between COVID-19 pandemic and Artificial Intelligence (AI): Can it be the catalyst for Society 5.0. Int J Sci Res Manag. 2020; 8: 2011-2025.

49. Albanesi S, Kim J. Effects of the COVID-19 recession on the US labor market: Occupation, family, and gender. J Econ Perspect. 2021; 35: 3-24.

50. Nguyen MH, Hunsaker A, Hargittai E. Older adults’ online social engagement and social capital: The moderating role of Internet skills. Inf Commun Soc. 2020: 1-17.

51. Arcury TA, Sandberg JC, Melius K P, et al. Older adult internet use and eHealth literacy. J Appl Gerontol. 2020; 39: 141-150.

52. Vahia VN, Shah AB. COVID-19 pandemic and mental health care of older adults in India. Int Psychogeriatr. 2020; 32: 1125-1127.

53. Qiu LJ, Zhong SB, Sun BW, et al. Is internet penetration narrowing the rural–urban income inequality? A cross-regional study of China. Qual Quant. 2021; 55: 1795-1814.

Thank you again for your suggestions to better improve the article.

Response to Editor

Point 1: We note that you have provided funding information that is not currently declared in your Funding Statement. However, funding information should not appear in the Acknowledgments section or other areas of your manuscript. We will only publish funding information present in the Funding Statement section of the online submission form. 

Please remove any funding-related text from the manuscript and let us know how you would like to update your Funding Statement. 

Response 1: Thank you for pointing out the improvement direction of my manuscript.

Funding-related text has been removed from the manuscript and this study received no external or internal funding, which has been amended within the cover letter.

Thank you again for your suggestions to improve the article.

Point 2: Please provide an amended statement that declares *all* the funding or sources of support (whether external or internal to your organization) received during this study, as detailed online in our guide for authors at http://journals.plos.org/plosone/s/submit-now. Please also include the statement “There was no additional external funding received for this study.” in your updated Funding Statement. 

Response 2: Thank you for pointing out this direction for the improvement of my manuscript.

I have amended the funding statement in the cover letter by declaring “There was no additional external or internal funding received for this study.”

Thank you again for your suggestions to improve the article.

Point 3: We note that you have stated that you will provide repository information for your data at acceptance. Should your manuscript be accepted for publication, we will hold it until you provide the relevant accession numbers or DOIs necessary to access your data. If you wish to make changes to your Data Availability statement, please describe these changes in your cover letter and we will update your Data Availability statement to reflect the information you provide.

Response 3: Thank you for pointing out this direction for the improvement of my manuscript.

Repository information for my data at acceptance is as follows: https://doi.org/10.7910/DVN/SZUSBS. And it could be accessed anytime from now.

Thank you again for your suggestions to improve the article.

---

## [Decision Letter · Decision Letter 1]

20 Jun 2022

PONE-D-22-00051R1The role of internet use in the relationship between occupational status and depressionPLOS ONE

Dear Dr. Zhang,

Thank you for submitting your manuscript to PLOS ONE. After careful consideration, we feel that it has merit but does not fully meet PLOS ONE’s publication criteria as it currently stands. Therefore, we invite you to submit a revised version of the manuscript that addresses the points raised during the review process.

We look forward to receiving your revised manuscript.

Kind regards,

László Vasa, PhD

Academic Editor

PLOS ONE

Reviewers' comments:

Reviewer's Responses to Questions

**Comments to the Author**

1. If the authors have adequately addressed your comments raised in a previous round of review and you feel that this manuscript is now acceptable for publication, you may indicate that here to bypass the “Comments to the Author” section, enter your conflict of interest statement in the “Confidential to Editor” section, and submit your "Accept" recommendation.

Reviewer #2: All comments have been addressed

2. Is the manuscript technically sound, and do the data support the conclusions?

Reviewer #2: Partly

3. Has the statistical analysis been performed appropriately and rigorously? 

Reviewer #2: Yes

4. Have the authors made all data underlying the findings in their manuscript fully available?

Reviewer #2: No

5. Is the manuscript presented in an intelligible fashion and written in standard English?

Reviewer #2: Yes

6. Review Comments to the Author

Reviewer #2: It can be seen that the authors have responded to all the points made by previous reviewers.

The introduction has been expanded and indeed numerous sources are cited. However, the nature of the topic also justifies a much more sophisticated, thematic literature review, which provides a better grounding in the theoretical background of the topic and the research model.

It is not only worthwhile to substantiate the topicality of the topic, but also to present, in an appropriate logical order, the relevant aspects of previous case studies, research and theoretical models that underpin the research and support its conceptualisation.

The methodology is improved and more detailed than in the previous version. However, the full process of data collection is still to be presented. How the data was collected, how the sample was taken and what code of ethics was followed. A more detailed description of the research tool is also recommended to better understand the methodological background of the results presented in the presentation of the research results. The Likert scale is a type of scale for assessing attitudes. I propose to justify why this type was chosen and, within that, to explain the rationale for the scale of 1-5, according to methodological and professional grounds. I suggest a more detailed explanation of the relationship between the research objective, the research hypothesis, the chosen statistical measures and the results. This will provide a methodological and logical basis for the results of the measurements presented in the research results and the novelty of the research. I suggest expanding the conclusions: a practical summary of the main findings of the research, a more detailed description of the limitations of the research and suggestions for further research. It would also be useful, in my view, if the authors were to include in this chapter their findings and their conclusions on the adaptation of the research results and methodology.

7. PLOS authors have the option to publish the peer review history of their article (what does this mean?). If published, this will include your full peer review and any attached files.

Reviewer #2: **Yes: **Mónika Garai-Fodor

---

## [Author Response · Author response to Decision Letter 1]

28 Jun 2022

Dear Editor and Reviewers,

Thank you for the opportunity to let me resubmit my revised manuscript, “The role of internet use in the relationship between occupational status and depression”, for publication in PLoS One.

In the resubmission, I have carefully responded to the suggestions in the attached Reviewers’ comments. Meanwhile, I modified the manuscript accordingly. When I read the comments, I was very grateful for the comments, which enabled me to rethink about the theoretical and practical implications, as well as the limitations of my manuscript. I have also reorganized the paragraphs according to these valuable suggestions.

I would like to express my deep appreciation for your response and suggestions. I hope that the revised manuscript is now acceptable for publication. I look forward to hearing from you soon.

Sincerely yours,

Yujie Zhang

Shanghai Jiao Tong University

No. 1954 Hua Shan Road

Shanghai 200030, China

Tel. +86 18696960193

Email. zhangyujie@sjtu.edu.cn

Response to Reviewer 2 Comments

Title & Abstract

Point 1: 1. If the authors have adequately addressed your comments raised in a previous round of review and you feel that this manuscript is now acceptable for publication, you may indicate that here to bypass the “Comments to the Author” section, enter your conflict of interest statement in the “Confidential to Editor” section, and submit your "Accept" recommendation.

Reviewer #2: All comments have been addressed

Response: Thank you for your important suggestions, which were very helpful for the manuscript improvement.

Thank you again for your suggestions to better improve the article.

Point 2: 2. Is the manuscript technically sound, and do the data support the conclusions?

Reviewer #2: Partly

Response: Thank you for your important suggestions, which were very helpful for the manuscript improvement.

The revised manuscript improved the data presentation and reorganized the conclusions. As follows:

Methods

Data and sample

The cross-sectional data of this study were obtained from the 2017 wave of the Chinese General Social Survey (https://doi.org/10.7910/DVN/SZUSBS). The CGSS was conducted by the Survey and Data Center of Renmin University of China. At present, CGSS data has become the main data source for the study of Chinese society and is widely used in scientific research, teaching, and government decision-making. CGSS adopted stratified multiple-stage probability sampling method in the survey, and was carried out in all urban and rural households in 31 provinces, autonomous regions, and municipalities directly under the central government (excluding Hong Kong, Macao, and Taiwan), which makes CFPS data representative and authoritative with scientific research value. Moreover, CFPS was implemented by a group of trained researchers through face-to-face interviews, later family visits, and telephone surveys, which ensures the high quality of the data. CGSS was conducted following the ethical principles in the Declaration of Helsinki. After ensuring that the potential respondents understood the information, the researchers obtained the independent informed consent of the respondents. Each potential respondent was fully informed of the research purpose, method, funding source, any possible conflict of interest, institutional affiliation of the researchers, expected benefits and potential risks of the research, possible discomfort caused by the research, and any other information related to the research.

The sample size of this project was 12,582. The subjects of this study were older adults over 60 years old. After the screening and elimination of samples lacking relevant variables, 2,403 effective samples were obtained.

Because the present study was conducted based on the de-identified, publicly available CGSS data, it does not constitute human subject research. Its institutional review board review was waived because there was no interaction with any individual, and no identifiable private information was used.

Measures

In this study, three core variables (i.e. occupational status, internet use and depression) were measured using the Likert 5-point scale. The Likert 5-point scale consists of a group of statements. Each statement has five responses, namely "strongly agree", "agree", "not necessarily", "disagree" and "strongly disagree", which are recorded as 5, 4, 3, 2 and 1 respectively. What is measured is the attitude or status of each respondent on the scale [52]. The Likert scale usually uses five response levels, but many psychometrists advocate using more levels [53]. However, a recent empirical study pointed out that after simple data conversion, the average, variance, skewness and kurtosis of the data with 5-level, 6-level and 11-level options are very similar, so statistically speaking, there is little difference [54]. In addition, although the literature shows that the Likert scale with more levels of options is more sensitive. Validity may also improve [55]. However, Chinese people are not very sensitive to the use of words, and some words are easy to confuse the respondents [56]. For example, "some agree" and "a little agree", some respondents reported that it is difficult to distinguish. In addition, the subjects of this study are older adults, who do not have a strong ability to distinguish the details of language words [57]. Therefore, the Likert 5-point scale is more suitable to measure their state.

Depression 

In CGSS, respondents were asked about their self-reported feeling of depression as follows: “How often have you felt depressed or frustrated in the past four weeks?” This item was scored on a 5-point Likert scale ranging from 1(never) to 5(always).

Occupational status

Research of occupational status in China suggested that in people’s subjective cognition, the Communist Party of China (CPC) and the government has the highest occupational status, enterprise takes the second place [58]. As there were wage restrictions for nonprofits and grass-roots organizations in China, and their legitimacy depended on relationships with the government, public institutions affiliated to government had higher occupational status compared with nonprofits or grass-roots organizations [59]. Since there was no social security or social recognition for unemployed individuals, the occupational status of unemployment was the lowest [60]. In CGSS, respondents were asked about their occupational status, namely, "What is your occupational status type?" (“unemployment” was coded as 1, “nonprofits or grass-roots organizations” was coded as 2, “public institutions” was coded as 3, “enterprise” was coded as 4, and “the Communist Party of China and the government” was coded as 5). In general, the higher the score, the higher the occupational status is. 

Internet use

In CGSS, respondents were asked about their internet use frequency, namely, “In the past year, do you often engage in online activities in your spare time?” (“never” was coded as 1, “several times a year or less” was coded as 2, “several times a month” was coded as 3, “several times a week” was coded as 4, and “everyday” was coded as 5).

Covariates

Based on the existing literature [55-58], age (continuous variable), sex (0 = “female”, 1 = “male”), education (1 = “below junior high school”, 2 = “above junior high school, and below undergraduate”, 3 = “above undergraduate”), income (1 = “below RMB 50,000 per year”, 2 ="above RMB 50,000, and below RMB 100,000 per year”, 3 = “above RMB 100,000 per year”), social class (1-10, with a higher score indicating a higher social class) and residence registration (0 = “rural”, 1 = “urban”) were selected as control variables.

Statistical analysis

All the statistical analyses were performed with Stata version 16.0 (StataCorp, Texas of United States). Stata is a statistical program developed by StataCorp in 1985. It is widely used in enterprises and academic institutions. Many users work in research fields, especially in economics, sociology, politics and epidemiology. Stata has a strong statistical function. In addition to the traditional statistical analysis methods, it also collects new methods developed in recent 20 years, such as Cox proportional risk regression, exponential and Weibull regression, logistic regression between multi category results and ordered results, Poisson regression, negative binomial regression and generalized negative binomial regression, random effect model, etc. Specifically, Stata has the following statistical analysis capabilities: general analysis of numerical variable data: parameter estimation, one-way and multi factor variance analysis, covariance analysis, interaction effect model, balanced and unbalanced design, nested design, random effect, pairwise comparison of multiple means, processing of missing data, variance homogeneity test, normality test, etc.

The statistical analysis is conducted in four stages. First, this study conducted a descriptive analysis of the variables. Next, based on an OLS model, a moderating effect analysis was performed to test whether internet use moderated the association between occupational status and depression. Then, based on residence registration, a heterogeneity analysis was performed to test if occupational status moderated the association between occupational status and depression both in rural and urban groups. Lastly, a robustness test was performed by using the ordered logit regression. And the results were robust. In the present study, the Variance Inflation Factor (VIF) values < 10, which indicated that multicollinearity was not an issue in the estimate. Figure 1 shows the research framework.

Thank you again for your suggestions to better improve the article.

Point 3: 3. Has the statistical analysis been performed appropriately and rigorously?

Reviewer #2: Yes

Response: Thank you for your important suggestions, which were very helpful for the manuscript improvement.

Thank you again for your suggestions to better improve the article.

Point 4: 4. Have the authors made all data underlying the findings in their manuscript fully available?

Reviewer #2: No

Response: Thank you for your important suggestions, which were very helpful for the manuscript improvement.

The data underlying the findings in this manuscript have been made fully available. As follows:

The cross-sectional data of this study were obtained from the 2017 wave of the Chinese General Social Survey (https://doi.org/10.7910/DVN/SZUSBS).

Thank you again for your suggestions to better improve the article.

Point 5: 5. Is the manuscript presented in an intelligible fashion and written in standard English?

Reviewer #2: Yes

Response: Thank you for your important suggestions, which were very helpful for the manuscript improvement.

Thank you again for your suggestions to better improve the article.

Point 6: 6. Review Comments to the Author

Reviewer #2: It can be seen that the authors have responded to all the points made by previous reviewers.

The introduction has been expanded and indeed numerous sources are cited. However, the nature of the topic also justifies a much more sophisticated, thematic literature review, which provides a better grounding in the theoretical background of the topic and the research model.

It is not only worthwhile to substantiate the topicality of the topic, but also to present, in an appropriate logical order, the relevant aspects of previous case studies, research and theoretical models that underpin the research and support its conceptualisation.

The methodology is improved and more detailed than in the previous version. However, the full process of data collection is still to be presented. How the data was collected, how the sample was taken and what code of ethics was followed. A more detailed description of the research tool is also recommended to better understand the methodological background of the results presented in the presentation of the research results. The Likert scale is a type of scale for assessing attitudes. I propose to justify why this type was chosen and, within that, to explain the rationale for the scale of 1-5, according to methodological and professional grounds. I suggest a more detailed explanation of the relationship between the research objective, the research hypothesis, the chosen statistical measures and the results. This will provide a methodological and logical basis for the results of the measurements presented in the research results and the novelty of the research. I suggest expanding the conclusions: a practical summary of the main findings of the research, a more detailed description of the limitations of the research and suggestions for further research. It would also be useful, in my view, if the authors were to include in this chapter their findings and their conclusions on the adaptation of the research results and methodology.

Response: Thank you for your important suggestions, which were very helpful for the manuscript improvement.

As the nature of the topic justifies a much more sophisticated, thematic literature review, the revised manuscript rewrote the literature review part in order to provide a better grounding in the theoretical background of the topic and the research model. Moreover, in addition to substantiate the topicality of the topic, the revised manuscript presented the literature review in an improved logical order. By including the relevant aspects of previous case studies, research and theoretical models that underpin the research and its conceptualisation, the revised literature review is used to better support this study. As follows:

Introduction

In most societies, social members are often divided into different classes formally or informally due to their different characteristics such as power [1], property [2], education [3], family [4], race [5], gender [6], age [7], and occupation [8], that is, there is a system that gives different social members different social status. Occupational stratification is a very key feature to distinguish social members [9, 10]. 

In today's industrialized society, occupational status often becomes an indicator of a person's status in society and affects people's class mobility [11]. Therefore, the change of occupational status has also become an important indicator to predict the direction and degree of social class change. Past studies have shown that social class change is related to depression [12, 13]. Entering an aging society, older population is a highly vulnerable group to depression [14]. It is very important to understand the mechanism of how occupational status associates with depression and the role of internet use plays between these two variables. This exploration not only helps to promote the theoretical research on social stratification and social mobility, but also has practical significance. Especially in the context of aging and information society, it helps to clarify the main obstacles to promoting the mental health of older adults and the direction of future efforts.

This study focuses on two dimensions of occupational status: social stratification and social mobility. The former refers to the stratification phenomenon of social members due to different possession of social resources [15], which can be regarded as the root cause of depression symptoms. The latter refers to the upward, downward or horizontal mobility of individuals in the social class [16]. Usually, social class is the main factor that can directly affect mental health [17], similar to proximate causes. Based on the review of existing literature and the theoretical analysis of social stratification and social mobility, this paper proposes that there is a positive relationship between occupational status and depression. Since entering the society 5.0 [18], where information technology has changed people's way of life, internet use has a moderating effect between occupational status and depression. More frequent internet use can significantly weaken the relationship between occupational status and depression. The empirical part focuses on China, using a national cross-sectional data set, and using statistical analysis methods to test the theoretical assumptions.

There are three main points of view about the impact of occupational status on depression. One view is that occupational status can promote mental health and alleviate depressive symptoms. Diaz et al. believed that low subjective occupational status perception was an important factor in predicting depression symptoms [19]. Murphy et al. found that the prevalence of depression in people with low occupational status was significantly higher than that in people with high occupational status. There is also a trend that the downward social mobility predicted by low occupational status is related to depression. This finding supports the view that depressed people are concentrated at the low end of the social class [20]. Singh-Manoux et al. analyzed the prospective cohort study of civil servants in London. It shows that subjective occupational status perception is a powerful predictor of poor mental health. 

Subjective occupational status reflects the average cognitive level of socio-economic status, without psychological bias [21]. In this regard, common explanations include that the social stratification system will produce differences in mental health [22], and the sense of unfairness caused by poor social mobility [23]. Generally speaking, this kind of view is mainly used to discuss the occupational status of western developed countries. The main characteristics of western developed countries are that the economic development speed slows down, the mobility of social classes decreases, the social class is relatively solidified, it is very difficult for the bottom to squeeze into the middle class, and it is more difficult for the middle class to squeeze into the top [24-26]. In developing countries like China, the economic development is in the stage of rapid and steady growth, and the proportion of the middle class is still at a relatively low level [27]. In the future, a considerable number of rural families or working families will squeeze into the middle class through efforts and ability. Some middle-class families with strong ability can even squeeze into the rich class. Nevertheless, this involves efforts, capabilities, networks and other aspects [28].

Another view is that occupational status will aggravate psychological stress and aggravate depressive symptoms. Demerouti et al. and Seto et al. used the methods of causal analysis of longitudinal data sets and correlation analysis of cross-sectional data sets to test the negative effects of work stress and family career conflict caused by high occupational status, and found that high occupational status can lead to the aggravation of depression, and work stress and fatigue are the determinants of depression; Moreover, as time goes on, work stress and fatigue have a causal relationship and a reverse causal relationship, thus forming a vicious circle [29, 30]. Song's comparative study and analysis of China and the United States found that, unlike the positive effect of high occupational status on mental health in the United States, in Chinese cities, the upward (relative to downward) social comparison motivation is stronger than that in the American society with a strong individualist culture. High occupational status also means facing higher peer pressures, which may cause individuals to feel dissatisfied with their own situation, thus aggravating depression symptoms [31].

The third view is that the relationship between occupational status and depression is regulated by other factors. For example, Nishimura pointed out that although it was found in the comparative study of Japan, South Korea and China that occupational status had an impact on depression, the investigation of the labor market structure of three different societies showed that the impact of occupational status was regulated by other variables, such as social culture and institutional framework. High occupational status can reduce depression symptoms through two distinct mechanisms: one is to provide sufficient financial stability and provide employees with additional benefits and other potential economic resources; the other is to avoid employees from experiencing various stressors corresponding to low occupational status, such as economic stress and work overload. In China, for example, people working in private enterprises have a higher level of depression than those working in the party and government departments, partly because of the greater economic pressure and work overload in private enterprises [32]. In addition, even for older adults, the soundness of the social welfare system also plays a regulatory role: for example, for low-income countries and middle-income countries, in China, where the soundness of social welfare is relatively strong, high occupational status means higher pensions, which to a large extent can alleviate the depressive symptoms of older adults; However, in India, where the social welfare system is relatively unsound, the dependence between social welfare and occupational status is weak, and it is difficult for occupational status to alleviate depressive symptoms through the welfare mechanism [33].

Based on the existing literature, this paper attempts to test the relationship between occupational status and depression. This paper mainly focuses on the occupational status of developing countries, and its association with depression is significantly different from that of western developed countries. First, the social stratification in western developed countries is relatively fixed, and occupation can be used as a representative standard for stratification. In developing countries, social mobility is fast, and the types of emerging occupations continue to iterate. The correlation between occupation status and depression may be weaker than that in western developed countries. Second, for developing countries, the rapid urbanization process has widened the gap between urban and rural areas. Because the rural vocational system is single, mainly focusing on agriculture, while the urban vocational system is diverse, forming a differentiated hierarchical system. Therefore, the association between occupational status and depression has urban-rural heterogeneity, which is more significant in cities. Third, different from the Chinese society before the reform and opening up, since the reform and opening up in 1978, the Chinese government has gradually established a modern social security system. For older population, high occupational status also means higher pension and more stable welfare security, which will promote their mental health after retirement [34]. Therefore, this study proposes the following hypothesis.

Hypothesis 1: Occupational status is negatively associated with depression. 

The relevant literature discusses the relationship between internet use, mental health and occupational status. On the one hand, scholars generally recognize that internet use is conducive to improving depression. For example, Liu et al. proposed that internet use is conducive to promoting social participation and has important value in maintaining mental health [35]; On the other hand, König et al. found that in Europe, the internet use of older adults is driven by the impact of previous work experience. Behind the differences in occupational status is the impact of social inequality and different social and economic resources on older population’s access to new technologies for internet use [36]. At the same time, older adults with low internet use skills are increasingly affected by the negative impact of social stratification. In the view of some scholars, it is necessary to focus on the social 5.0 perspective and help the older adults improve their mental health by providing internet use training and other means to resist the negative impact of social stratum consolidation and poor mobility [37-39].

In recent years, more scholars have tested the role of internet use in social stratification and social mobility based on empirical research design. Eynon et al. based on the analysis of four waves of longitudinal data from the United Kingdom, found that internet use helps to improve social mobility, and this effect is more significant in western developed countries and the promotion of social stratification represented by occupational status [40]. Kappeler et al. took how the digital divide in Switzerland - social differences in Internet adoption - evolved from 2011 to 2019 as an example, and found that the use of the Internet was still stratified according to existing social stratifications, while the situation of not using the Internet was increasingly concentrated in traditionally disadvantaged occupations [41]. Yoon et al. found that for older adults, the lower level of occupational status significantly reduced the probability of using the Internet to obtain health information [42]. Barbosa Neves et al. proposed that the Internet seems to help maintain, accumulate and even mobilize social capital. Moreover, it also seems to exacerbate social inequality and cumulative advantage [43]. Based on Max Weber's political perspective, Blank and Groselj pointed out that the main sources of social stratification are social class, occupational status and power. As the Internet becomes more and more important, it has steadily risen to a more central position in the hierarchical system. Therefore, it is important to look at the use of the Internet from the perspective of Marx Weber, and explore how social class, occupational status and power help explain people's Internet realization [44]. In addition, although recent studies have shown that the scale of the digital divide has been shrinking for many groups, there is a strong correlation between socio-economic factors such as occupational status and internet use [45].

Based on the literature, this study further points out that internet use plays an important role in reducing the relationship between occupational status and depression. The existing literature mainly discusses the impact of internet use from the perspective of social stratification and social mobility. However, the association between occupational status and depression of older adults is long-term and has accumulated for many years. Some older adults have even retired, so it is difficult to change their occupational status. Will the use of the Internet help ease this long-term relationship? At present, there is still a lack of detailed demonstration on this issue, but this paper believes that the answer is yes. Internet use has a positive effect on alleviating the long-term relationship between occupational status and depression.

Internet use is a multi-dimensional concept. This study focuses on the role of internet use frequency. The frequency of internet use has a direct impact on mental health, which determines whether individuals can effectively participate in society and accumulate social capital in the information age. It is a direct factor affecting mental health. Choi and DiNitto found that internet use can establish social integration and support networks for older adults, and enhance psychological capital, namely emotional well-being and self-efficacy, based on a representative sample of the older population aged 65 and over in the United States [46]. Older people who use the Internet more often usually report better self-rated health [47]. According to Matthews et al., for the older adults with lower socio-economic status, the increase of internet use frequency has greater health benefits than those with higher socio-economic status [48]. In the long run, because the use of the Internet has improved social capital, it has also reduced the older population's concern that their social and economic status is inferior to others to a certain extent [49]. In reality, the low occupational status often restricts the improvement of the economic ability of older adults, so that too many resources benefit the people with high occupational status. The living conditions of people with low occupational status are relatively poor, which is not conducive to mental health [50]. However, the increase of social capital caused by internet use improves the social support that older adults can obtain. Social support is conducive to mental health [51]. Therefore, internet use is an inhibitory intermediary factor between occupational status and depression. This study proposes the following hypothesis.

Hypothesis 2: Internet use is an inhibitory moderator between occupational status and depression.

To the best of the author’s knowledge and based on this research, there were no previous studies concerning the relationship between occupational status, internet use and depression among older adults in China. This study was conducted to fulfill this research gap. The purpose of this study was to estimate the moderating role of internet use on the association between occupational status and depression in society 5.0. 

The full process of data collection is presented in the revised manuscript to show how the data was collected, how the sample was taken and what code of ethics was followed. As follows:

Data and sample

The cross-sectional data of this study were obtained from the 2017 wave of the Chinese General Social Survey (https://doi.org/10.7910/DVN/SZUSBS). The CGSS was conducted by the Survey and Data Center of Renmin University of China. At present, CGSS data has become the main data source for the study of Chinese society and is widely used in scientific research, teaching, and government decision-making. CGSS adopted stratified multiple-stage probability sampling method in the survey, and was carried out in all urban and rural households in 31 provinces, autonomous regions, and municipalities directly under the central government (excluding Hong Kong, Macao, and Taiwan), which makes CFPS data representative and authoritative with scientific research value. Moreover, CFPS was implemented by a group of trained researchers through face-to-face interviews, later family visits, and telephone surveys, which ensures the high quality of the data. CGSS was conducted following the ethical principles in the Declaration of Helsinki. After ensuring that the potential respondents understood the information, the researchers obtained the independent informed consent of the respondents. Each potential respondent was fully informed of the research purpose, method, funding source, any possible conflict of interest, institutional affiliation of the researchers, expected benefits and potential risks of the research, possible discomfort caused by the research, and any other information related to the research.

The sample size of this project was 12,582. The subjects of this study were older adults over 60 years old. After the screening and elimination of samples lacking relevant variables, 2,403 effective samples were obtained.

Because the present study was conducted based on the de-identified, publicly available CGSS data, it does not constitute human subject research. Its institutional review board review was waived because there was no interaction with any individual, and no identifiable private information was used.

Besides, a more detailed description of the research tool is added to better understand the methodological background of the results presented in the results part. As follows:

Statistical analysis

All the statistical analyses were performed with Stata version 16.0 (StataCorp, Texas of United States). Stata is a statistical program developed by StataCorp in 1985. It is widely used in enterprises and academic institutions. Many users work in research fields, especially in economics, sociology, politics and epidemiology. Stata has a strong statistical function. In addition to the traditional statistical analysis methods, it also collects new methods developed in recent 20 years, such as Cox proportional risk regression, exponential and Weibull regression, logistic regression between multi category results and ordered results, Poisson regression, negative binomial regression and generalized negative binomial regression, random effect model, etc. Specifically, Stata has the following statistical analysis capabilities: general analysis of numerical variable data: parameter estimation, one-way and multi factor variance analysis, covariance analysis, interaction effect model, balanced and unbalanced design, nested design, random effect, pairwise comparison of multiple means, processing of missing data, variance homogeneity test, normality test, etc.

The statistical analysis is conducted in four stages. First, this study conducted a descriptive analysis of the variables. Next, based on an OLS model, a moderating effect analysis was performed to test whether internet use moderated the association between occupational status and depression. Then, based on residence registration, a heterogeneity analysis was performed to test if occupational status moderated the association between occupational status and depression both in rural and urban groups. Lastly, a robustness test was performed by using the ordered logit regression. And the results were robust. In the present study, the Variance Inflation Factor (VIF) values < 10, which indicated that multicollinearity was not an issue in the estimate. Figure 1 shows the research framework.

A justification is added to explain why Likert scale was chosen and the rationale for the scale of 1-5, according to methodological and professional grounds. As follows:

Measures

In this study, three core variables (i.e. occupational status, internet use and depression) were measured using the Likert 5-point scale. The Likert 5-point scale consists of a group of statements. Each statement has five responses, namely "strongly agree", "agree", "not necessarily", "disagree" and "strongly disagree", which are recorded as 5, 4, 3, 2 and 1 respectively. What is measured is the attitude or status of each respondent on the scale [52]. The Likert scale usually uses five response levels, but many psychometrists advocate using more levels [53]. However, a recent empirical study pointed out that after simple data conversion, the average, variance, skewness and kurtosis of the data with 5-level, 6-level and 11-level options are very similar, so statistically speaking, there is little difference [54]. In addition, although the literature shows that the Likert scale with more levels of options is more sensitive. Validity may also improve [55]. However, Chinese people are not very sensitive to the use of words, and some words are easy to confuse the respondents [56]. For example, "some agree" and "a little agree", some respondents reported that it is difficult to distinguish. In addition, the subjects of this study are older adults, who do not have a strong ability to distinguish the details of language words [57]. Therefore, the Likert 5-point scale is more suitable to measure their state.

A more detailed explanation of the relationship between the research objective, the research hypothesis, the chosen statistical measures and the results, so as to provide a methodological and logical basis for the results of the measurements presented in the research results and the novelty of the research. As follows:

This study began with the theory of social stratification and social mobility, added the theory of social capital, constructed an analytical framework of regulatory effect, put forward two basic assumptions, selected measurement indicators taking into account Chinese cultural factors, and conducted empirical tests using authoritative statistical software and national representative data, providing a new theoretical contribution to our understanding of the impact of occupational status and depression in developing countries.

The conclusions were expanded as follows: a practical summary of the main findings of the research, a more detailed description of the limitations of the research and suggestions for further research. Moreover, the findings and conclusions on the adaptation of the research results and methodology were added.

Discussion

This study attempts to explain whether occupational status is related to depression and the regulatory role of internet use in the above relationship. The results showed that: in general, the occupational status was indeed negatively correlated with depression, and the depressive symptoms of people with higher occupational status were lighter; Internet use has a negative regulatory effect on the relationship between occupational status and depression. High frequency internet use helps to alleviate the negative correlation between occupational status and depression.

This study has clear policy implications for promoting mental health and alleviating the negative psychological effects of the solidification of social stratification and the slowdown of social mobility. First, we should make effective use of digital technology empowerment to bridge the digital divide and promote the construction of Internet skills training programs for older adults. Internet application is an important symbol of social 5.0 and an important way to accumulate social capital [43, 46]. Targeted promotion projects must be designed to bridge the long-term negative psychological effects caused by low professional status [22-25]. At present, there is a huge digital divide in society. Older adults with low professional status have weak internet use skills [44]. The realization of the positive role of the Internet needs to be based on sufficient infrastructure construction, that is, the government needs to strengthen the Internet infrastructure construction first. Therefore, we should give further play to the enabling role of digital technology in mental health, promote the construction of society 5.0 nationwide, especially in rural areas with low Internet penetration, and break the phenomenon of "digital divide" [42, 45]. At the same time, we should establish a smooth social flow channel and a sound social security system, so that even older adults with low professional status can have a better mental health status and get equal social service resources.

This study has several limitations. First, the study has limited generalizability. Although CGSS encompasses a large range of older adults, respondents in this study are above the age of 60. This study uses age 65 as the lower threshold for older adults even though 60 is relatively old in the context of life expectancies in developing countries. Therefore, setting the cutoff to an older age limited the sample too much for generalization of the conclusions in developing countries. Future research is needed to explore the relationship using nationally representative data for individuals before age 60.

Second, this study was not able to examine reasons for the selection of occupation and how those factors might interact with mental health of older adults. Factors that related to the selection of occupations (e.g., social connections, technical skills) may impact mental health differently, and those effects may differ by occupational status of older adults. Therefore, qualitative studies of this topic could be carried out to reveal the nuanced mechanism that links occupational status and depression.

Finally, the results were cross-sectional; thus, they only examined the relationship between occupational status, internet use, and depression. Longitudinal and experimental studies should be conducted to reveal the causal relationship among the above variables.

The strengths of this article are related to the straightforward measurement of occupational status, internet use and depression using Likert 5-point scale and the large number of respondents that are studied. First, the way three core variables are measured is an improvement upon many other studies. In line with theoretical considerations of the applicability, three core variables are measured by questions regarding the state of the respondents [52, 53]. Second, based on the cultural background of China, this study measured three core variables using 5 levels of Likert scale [56], which resulted in reliable estimations of the respondents’ status. Third, this study systematically accounted for individual demographic and socio-economic conditions as well, while studying the relationship between occupational status, internet use and depression.

The findings here are in line with the earlier results of Choi and DiNitto, who also established a conceptual framework to investigate how internet use could accumulate social capital to alleviate depression of older adults by improving self-efficacy [46]. Although social stratification theory and social mobility theory already provide a good introduction to how occupational status is associated with depression [16-19], the perspective of social capital theory provides a new discovery for us to understand how internet use weakens this connection. In this case, this study began with the theory of social stratification and social mobility, added the theory of social capital, constructed an analytical framework of regulatory effect, put forward two basic assumptions, selected measurement indicators taking into account Chinese cultural factors, and conducted empirical tests using authoritative statistical software and national representative data, providing a new theoretical contribution to our understanding of the impact of occupational status and depression in developing countries.

Conclusion

This study examines the correlation between occupational status and depression and the role of internet use in moderating the above relationship. This study found that occupational status was significantly negatively correlated with depression. An increase in internet use can inhibit the negative relationship between occupational status and depression.

Thank you again for your suggestions to better improve the article.

---

## [Editor Report · Decision Letter 2]

25 Jul 2022

The role of internet use in the relationship between occupational status and depression

PONE-D-22-00051R2

Dear Dr. Zhang,

We’re pleased to inform you that your manuscript has been judged scientifically suitable for publication and will be formally accepted for publication once it meets all outstanding technical requirements.

Kind regards,

Prof. László Vasa, PhD

Academic Editor

PLOS ONE
---

## [Editor Report · Acceptance letter]

27 Jul 2022

PONE-D-22-00051R2 

The role of internet use in the relationship between occupational status and depression 

Dear Dr. Zhang:

I'm pleased to inform you that your manuscript has been deemed suitable for publication in PLOS ONE. Congratulations! Your manuscript is now with our production department. 

Kind regards, 

on behalf of

Prof. Dr. László Vasa 

Academic Editor

PLOS ONE